# Phytase and phosphate-solubilizing *Bacteria* addition improves P availability and uptake by maize in low-phosphorus red soil

Long Zhou [ID]¤, Litao Zhu¤, Faming Zhang [ID]¤*

College of Rural Revitalizing and Education of Yunnan, Yunnan Open University, Kunming, China

¤ Current address: No. 113, Xuefu Road, Wuhua District, Kunming City, Yunnan Province, China
* famingzhanga@163.com

## Abstract

This study investigated the effects of phytase (PHY) and phosphate-solubilizing *Bacillus* (PSB) isolates on maize growth and phosphorus (P) acquisition in low-P red soil, under conditions with or without organic P application. A pot experiment demonstrated that the combined application of organic P with exogenous PHY and PSB significantly enhanced maize agronomic traits, root morphology, rhizosphere properties, and P mobilization. Compared to the P-free control, organic P application alone increased plant height, stem diameter, dry matter, and P uptake by 9.9%, 5.4%, 23.2%, and 34.6%, root surface area, volume, and branching increased by 39.7%, 26.7%, and 44.7%, while average root diameter decreased by 19.5%, respectively. Exogenous phytase treatment was more effective than bacterial adding in enhancing P activation and uptake, increasing maize P absorption by 55.4% and 47.3% with and without sodium phytate, respectively. Redundancy analysis indicated positive correlations between root traits (root tips, surface area, volume) and P uptake, and between Olsen-P, resin-P, $NaHCO_3$-Pi, and enzyme activities. The results suggest that combining organic P with exogenous phytase effectively mobilizes native soil P pools, improves maize growth, and increases P use efficiency, offering a sustainable strategy for P management in low-P red soils.

## 1. Introduction

Phosphorus (P) is an essential macronutrient for crop growth and a major limiting factor for agricultural productivity, particularly in the acidic red soils of southern China [1,2]. In these highly weathered soils, applied P readily reacts with iron and aluminum oxides to form insoluble precipitates, resulting in 75–90% of fertilizer P being fixed and only 10–25% becoming available for plant uptake in the growing season [3]. Despite this low efficiency, global P fertilizer application has increased considerably due to rising food demand [4,5], leading to a 10% annual increase in P surplus from 1980 to 2018 [6,7]. This overuse not depletes finite, non-renewable phosphate rock

**Data availability statement:** All relevant data are within the manuscript and its Supporting information files.

**Funding:** This study received financial support from the Scientific Research Fund Project of Education Department of Yunnan Province in the form of a grant awarded to LZ (2026J0837). This study received additional financial support from the Scientific Research Fund Project of Education Department of Yunnan Province in the form of a grant awarded to FZ (2025J0731). No additional external funding was received for this study. The funders had no role in study design, data collection and analysis, decision to publish, or preparation of the manuscript.

**Competing interests:** The authors have declared that no competing interests exist. I have read the journal's policy and the authors of this manuscript have the following competing interests.

reserves—which may be exhausted within 50–100 years—but also poses serious environmental risks [8,9]. Therefore, improving P availability by reducing fixation and enhancing the mobilization of non-labile soil P pools is critical for sustainable agriculture and resource conservation [6–8].

Phosphorus-solubilizing microorganisms (PSMs) are key drivers of the soil P cycle and play a vital role in transforming both organic and inorganic P, enhancing the bioavailability of otherwise inaccessible P forms and promoting plant nutrient uptake [1,10]. Representative genera such as *Bacillus*, *Pseudomonas*, and *Rhizobium* enhance P availability by secreting organic acids (e.g., gluconic, acetic, and oxalic acids) that solubilize mineral phosphates, and extracellular phosphatases that mineralize organic P [11–15]. These microorganisms activate insoluble inorganic P through acidification (e.g., $H^+$ release) and carboxylate excretion, while also mineralizing organic P via increased phosphatase and phytase activities [16–18]. Inoculation with PSMs, especially when combined with organic amendments or insoluble P sources, has been shown to improve crop P uptake and yield [19–21]. For instance, Ramesh et al. demonstrated that PSB inoculation significantly increased soil available P and improved rice growth [22]. Similarly, Chang et al. [20] reported that PSB inoculation increased the labile P pool by 9.2% on average, while moderately labile and non-labile P pools decreased by 6.9% and 5.4%, respectively. However, most studies focus on plant responses and general soil properties, with limited attention to the dynamics and mechanisms of P pool transformation [18,23,24].

Microbial phosphatases dominate phosphatase activity in soils and significantly contribute to P bioavailability [25–28]. Notably, phytase—a subtype of phosphatase—catalyzes the hydrolysis of phytic acid, which constitutes 40–80% of soil organic P, into inorganic phosphate and inositol. It is estimated that phosphatases can hydrolyze up to 60% of soil organic P, with phytase playing a prominent role [10]. Exogenous phytase application has been demonstrated to enhance phosphatase activity, increase available P, and improve plant growth in neutral and alkaline soils [29–32]. Nevertheless, its effect in acidic red soils, particularly in combination with PSB and organic P sources, remains poorly understood.

Red soils account for approximately 20% of China's land area and support 40% of its population [33]. These soils are characterized by strong acidity, high weathering, and significant P fixation capacity due to intense desilication and aluminization [34]. As a result, the seasonal utilization efficiency of P fertilizer is often lower than 10% [35], making it imperative to develop strategies that improve P availability in these agronomically critical regions.

Plants adapt to P deficiency through various rhizosphere processes, including acidification, organic anion exudation, and phosphatase secretion [36–38]. The Hedley sequential fractionation method provides a detailed characterization of soil P pools, classifying them into labile (e.g., resin-P, $NaHCO_3$-Pi/Po), moderately labile (e.g., NaOH-Pi/Po, dilute HCl-Pi), and non-labile (concentrated HCl-Pi/Po, residual-P) forms [39,40]. This method offers insights into P turnover and availability following microbial or enzymatic interventions.

Although the combined application of PSMs and phytase has shown promise in increasing P uptake and crop yields [41–43], few studies have systematically examined their impact on P speciation and transformation in maize rhizosphere soil under organic P amendment. In particular, the interactions between exogenous phytase, bacterial inoculants, and native soil P pools in acidic red soils are not well documented.

This study investigates the effects of exogenous phytase (PHY) phosphate-solubilizing bacteria (*Bacillus velezensis*) (PSB) on P activation and maize P uptake in low-P red soil amended with organic P (sodium phytate). The specific objectives were to: (1) evaluate the effects of exogenous PSB and PHY on maize growth and P uptake under different P regimes; (2) to assess their influences on rhizosphere properties, soil physicochemical indicators, and P fractions following sodium phytate application; and (3) to elucidate the mechanisms through which these treatments mediate P activation and uptake in maize grown in low-P red soil.

## 2. Materials and methods

### 2.1. Preparation of experimental materials

A pot experiment was conducted in 2021 in a greenhouse at Yunnan agricultural university. The soil was collected from a dry red soil experimental site in Xiaoshao village, Guandu district, Kunming city, Yunnan province (24°54′N, 102°41′E; altitude 1820.0 m). The region has a mean annual temperature of 14.4 °C and receives an average annual rainfall of 850.0 mm. The soil is classified as mountainous red soil, characterized by strong acidity and low fertility. Key physicochemical properties included: pH 4.67, total P 0.22 g·kg$^{-1}$, soil organic matter 5.58 g·kg$^{-1}$, and Olsen-P 4.73 mg·kg$^{-1}$. The maize (*Zea mays* L.) cultivar used in the experiment was 'Yunrui 88'.

Phytase was obtained from Shanghai Yuanye Biotechnology Co., Ltd. The phosphate-solubilizing bacterial strain employed was *Bacillus velezensis* (GenBank Accession No. MW663765), provided by the Institute of agricultural resources and environment, Yunnan academy of agricultural sciences.

The phytase solution was prepared by dissolving solid phytase powder in sterilized distilled water under shaking to ensure homogeneity. For bacterial inoculation, a single colony of B. *velezensis* from a −80 °C stock was cultured in 10 mL of LB liquid medium (containing 10 g·L$^{-1}$ tryptone, 5 g·L$^{-1}$ yeast extract, and 10 g·L$^{-1}$ NaCl) at 30 °C with shaking at 180 rpm for 12–16 hours. Then, 2 mL of this pre-culture was transferred into 100 mL of fresh LB medium and incubated under the same conditions for 24 hours to prepare a high-concentration bacterial suspension. This seed culture was further expanded in 1 L of LB medium at a 5% (v/v) inoculation rate and incubated for 48 hours at 30 °C and 180 rpm. Bacterial cells were harvested by centrifugation at 8000 rpm for 10 min, washed twice with sterile saline solution (0.85% NaCl), and finally resuspended in sterile saline to achieve a concentration of approximately $10^{10}$ CFU·mL$^{-1}$ for soil inoculation.

The applied nutrient rates were 150 mg N·kg$^{-1}$, 100 mg $P_2O_5$·kg$^{-1}$, and 150 mg $K_2O$·kg$^{-1}$ soil. Nitrogen was supplied as urea (46% N), phosphorus as sodium phytate (46% $P_2O_5$), and potassium as potassium sulfate (51% $K_2O$). The nitrogen and potassium fertilizers were procured from Yunnan Yuntianhua Co., Ltd., while the sodium phytate was purchased from Shanghai Yuanye Biotechnology Co., Ltd.

### 2.2. Experimental design

The experiment was arranged in a randomized complete block design with three factors: (1) two organic P levels (with or without sodium phytate application); (2) two enzyme application levels (with or without phytase); and (3) two bacterial inoculation levels (inoculated with or without *B. velezensis*). This resulted in a total of 6 treatments, each replicated 4 times (24 pots in total). Each pot (18.0 cm × 11.0 cm × 12.5 cm) was filled with 2 kg of air-dried red soil sieved through a 2 mm mesh. Basal fertilizers were applied once at rates of 150 mg N·kg$^{-1}$, 100 mg $P_2O_5$·kg$^{-1}$, and 150 mg $K_2O$·kg$^{-1}$ soil. After watering, 3–4 maize seeds were sown per pot at a depth of 3–5 cm. Watering was performed every two days

to maintain adequate soil moisture. Approximately one week after emergence, two uniformly growing seedlings were retained per pot, and treatments involving exogenous PHY and PSB were initiated.

For phytase application, a total of 500 U per pot was applied in five split applications at 4-day intervals, each time using 50 mL of enzyme solution mixed with irrigation water. Control (CK) pots received an equal volume of water without enzyme. Bacterial inoculation was performed by applying 50 mL of B. *velezensis* suspension ($\approx 10^8$ CFU·mL$^{-1}$) evenly to the soil surface on the day of treatment, ensuring penetration into the rhizosphere zone. Standard agronomic management was maintained throughout the experiment, with no additional interventions aside from watering and thinning.

## 2.3. Sample collection and analysis

Plant sampling: Destructive plant harvesting was conducted 45 days after sowing (V12 stage). Plant height and stem diameter were measured using a tape measure and vernier caliper, respectively. Aboveground biomass and roots were carefully rinsed with distilled water, oven-dried at 105 °C for 30 minutes, then at 75 °C until constant weight, and dry weight was recorded. The entire root system was carefully excavated, and loosely adhered soil was removed. Roots were washed, collected over a 2 mm sieve, and stored at –20 °C for morphological analysis. Subsequently, roots were dried and used for total P determination. Plant and root P concentrations were analyzed via $H_2SO_4$–$H_2O_2$ digestion followed by the vanadomolybdate yellow colorimetric method [44].

Root morphology: Fresh roots were scanned using a digital scanner (Epson Expression1600 pro, Japan) at 720 dpi resolution. Root morphological traits including length, volume, surface area, average diameter, and number of root tips were quantified using WinRHIZO software (Version 4.0b, Regent Instruments Inc., Canada).

Soil sampling: Rhizosphere soil was collected using the shaking-root method. Visible root fragments were removed, and the soil was sieved (2 mm). One subsample was stored at –20 °C for enzyme activity assays, and another was air-dried and ground to pass through a 1 mm sieve for analysis of physicochemical properties and P fractions.

Soil analyses: Soil pH was measured potentiometrically in a 1:2.5 (w/v) soil–water suspension, soil organic matter (SOM) was determined using the potassium dichromate oxidation method, Olsen-P was extracted with 0.5 M NaHCO$_3$ (pH 8.5) and quantified by the ascorbic acid–molybdenum blue method, total P was measured after sulfuric–perchloric acid digestion followed by molybdenum blue colorimetry, available potassium was extracted with 1 M ammonium acetate and measured via flame photometry, alkali-hydrolyzable nitrogen was determined by the alkaline diffusion method [44].

Enzyme activities: Acid phosphatase (ACP) and alkaline phosphatase (ALP) activities were determined according to [25] by quantifying p-nitrophenol release from p-nitrophenyl phosphate. Briefly, 1 g of fresh soil was treated with toluene, modified universal buffer (pH 6.5 for ACP; pH 11.0 for ALP), and substrate, incubated at 37 °C for 1 h, then reacted with CaCl$_2$ and NaOH. Absorbance was read at 410 nm. Activities were expressed as µg *p*-nitrophenol released per gram dry soil per hour. Phytase activity was measured using the ammonium molybdate–vanadate yellow method as described by [10] with modifications from [40].

Mycorrhizal colonization: Mycorrhizal colonization is carried out using the microscopic staining examination method, the extent of arbuscular mycorrhizal fungal (AMF) colonization was assessed using 30 randomly selected root segments per sample, stained following [45], and observed under a light microscope. Colonization rates were quantified according to [46].

Soil P fractionation: Sequential extraction of soil P fractions was performed using the Hedley fractionation procedure as modified by [39]. P fractions included: Rresin-P, NaHCO$_3$-Pi, NaHCO$_3$-Po, NaOH-Pi, NaOH-Po, 1M HCl-Pi, conc. HCl-Pi, conc. HCl-Po, and residual-P. Based on bioavailability, these were grouped into labile P pool (Resin-P, NaHCO$_3$-Pi, NaHCO$_3$-Po), moderately labile P pool (NaOH-Pi, NaOH-Po, 1M HCl-Pi) and non-labile P pool (conc. HCl-Pi, conc. HCl-Po, residual-P) [45].

## 2.4. Data processing and statistical analysis

Data were processed using Microsoft Excel 2010, and graphs were generated with Origin 2018. Statistical analyses were performed using IBM SPSS Statistics (version 24.0; SPSS Inc., USA). One-way analysis of variance (ANOVA) was

applied to evaluate significant differences among treatment means. Redundancy analysis (RDA) was conducted using CANOCO 5.0 to explore relationships between environmental variables and soil P fractions [47].

## 3. Results

### 3.1. Plant growth and P uptake

The application of organic P significantly influenced maize growth and P acquisition, as reflected in plant height, stem diameter, dry matter weight, and P uptake (Fig 1). When sodium phytate was applied, maize plant height under the CK, PHY, and PSB treatments increased significantly by 8.8%, 11.1%, and 9.6%, respectively, compared to the corresponding treatments without sodium phytate. P uptake also rose significantly by 48.6%, 40.9%, and 14.4%. Additionally, dry matter under CK and PHY treatments increased significantly by 28.6% and 35.1%, respectively.

In the absence of sodium phytate, both PHY and PSB treatments resulted in significant improvements compared to CK: plant height increased by 8.5% and 4.5%, dry matter by 35.7% and 61.9%, and P uptake by 55.4% and 50.0%, respectively. Under sodium phytate amendment, the PHY treatment significantly enhanced plant height, dry matter, and P uptake by 10.7%, 42.6%, and 47.3% compared to CK. Similarly, the PSB treatment led to significant increases in dry matter weight and P uptake by 33.3% and 15.5%, respectively, relative to CK (Fig 1).

### 3.2. Root characteristics

The application of organic P, along with exogenous PSB and PHY, significantly influenced maize root morphological traits (Table 1). Compared to the no-P treatment, organic P application increased root surface area, root volume, and root branch number by 39.7%, 26.7%, and 44.7%, respectively, while reducing average root diameter by 19.5%. Among the treatments with organic P, the CK group showed significant increases in root length and root branch number by 55.1% and 19.8%, respectively. The PHY treatment led to significant increases in root length, root surface area, root tip number, root branch number, and root crossings by 63.8%, 92.4%, 102.6%, 98.2%, and 39.0%, respectively, along with a significant decrease in average root diameter of 23.3%. The PSB treatment resulted in a significant increase in root branch number by 16.1%, with no significant differences observed in other root traits.

In the absence of sodium phytate, the PHY treatment significantly enhanced root length, surface area, volume, tip number, branch number, and crossings by 112.2%, 70.6%, 67.0%, 73.9%, 100.8%, and 142.0%, respectively, compared to CK, while average root diameter decreased by 30.7%. The PSB treatment significantly increased root length, branch number, and crossings by 106.9%, 40.6%, and 233.7%, respectively, and significantly reduced average root diameter by 34.5%. Under sodium phytate application, the PHY treatment significantly improved root length, surface area, volume, tip number, branch number, and crossings by 124.2%, 176.9%, 98.9%, 369.4%, 232.3%, and 161.0%, respectively, relative to CK, and significantly decreased average root diameter by 20.9%. The PSB treatment significantly increased root length, tip number, branch number, and crossings by 23.8%, 48.2%, 36.3%, and 127.2%, respectively.

### 3.3. Soil physical and chemical properties

Compared to the no-P treatment, the application of organic P significantly influence several physicochemical properties of maize rhizosphere soil (Fig 2). Among them, the CK group showed significant increases in organic matter, available nitrogen, and available potassium by 15.5%, 48.6%, and 20.9%, respectively. The PHY treatment led to significant rises in Olsen-P and available potassium by 11.5% and 19.9%. Meanwhile, the PSB treatment significantly enhanced organic matter, Olsen-P, available nitrogen, and available potassium by 7.7%, 17.6%, 27.7%, and 25.9%, respectively. No significant differences were observed in the remaining indicators.

In the absence of sodium phytate, the PHY treatment significantly increased organic matter, Olsen-P, and available potassium by 18.4%, 85.7%, and 28.0%, respectively, compared to CK (Fig 2). Similarly, the PSB treatment resulted in

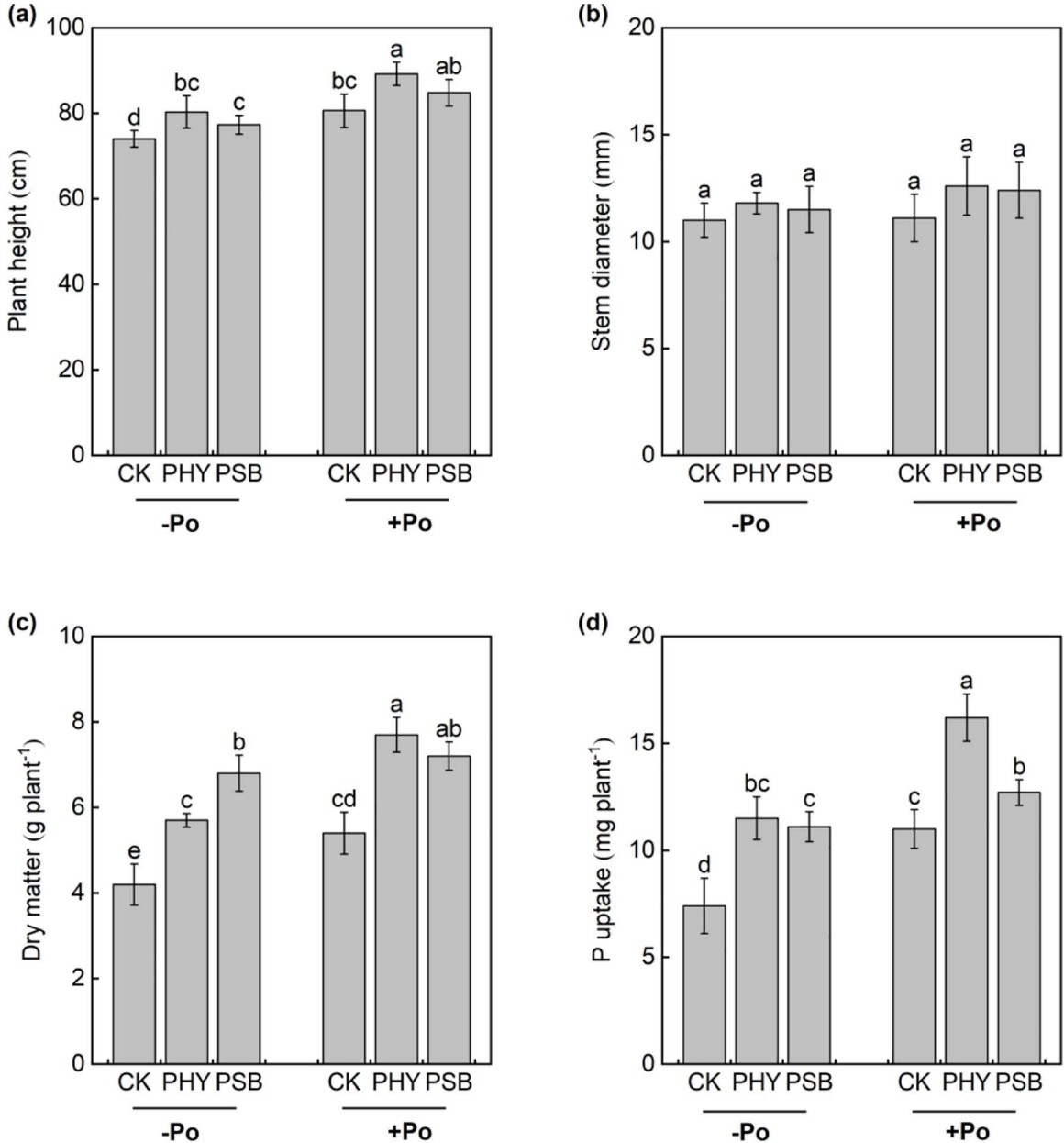

**Fig 1. Effects of exogenous addition with PHY and PSB on plant height (a), stem diameter (b), dry matter (c) and P uptake (d) of maize with or without application organic P in low-P red soil.** Values with the same lower-case letters are not significantly different among different P sources and exogenous addition treatments at the 5% level by the LSD. -Po represents that no organic P was applied, Po represents that organic P was applied. CK is control, PHY represents addition of phytase, PSB represents addition of phosphorus-solubilizing bacteria. the same as below.

significant elevations in organic matter, Olsen-P, and available potassium by 13.6%, 47.8%, and 15.2%. Under sodium phytate application, the PHY treatment significantly improved organic matter, Olsen-P, available nitrogen, and available potassium by 10.1%, 16.7%, 30.1%, and 27.0% relative to CK. The PSB treatment significantly enhanced soil organic matter (SOM), Olsen-P, available nitrogen, and available potassium by 5.9%, 11.1%, 26.9%, and 20.0%, respectively, while significantly reducing pH by 5.1% (Fig 2).

**Table 1. Effects of exogenous addition with PHY and PSB on root morphological traits of maize with or without application organic P in low-P red soil.**

| Treatments | | Root length (cm) | Root surface (cm²) | Root volume (cm³) | Average diameter (mm) | Number of root tips | Number of branches | Crossing number |
|---|---|---|---|---|---|---|---|---|
| -Po | CK | 917.8±210.7d | 139±25.2c | 1.29±0.26b | 0.53±0.01a | 4127±880c | 5640±894f | 419±75c |
| | PHY | 1947.6±121.6b | 237.2±19.2b | 2.15±0.41a | 0.37±0.04b | 7178±442b | 11323±205b | 1014±20b |
| | PSB | 1898.6±0b | 151.5±0c | 0.97±0b | 0.35±0.07bc | 4720±0c | 7930±174d | 1398±0a |
| +Po | CK | 1423.2±251.9c | 164.8±19.5c | 1.33±0.24b | 0.35±0.04b | 3099±555d | 6756±484e | 540±48c |
| | PHY | 3190.7±269.1a | 456.4±48.1a | 2.65±0.69a | 0.28±0.05c | 14545±650a | 22448±970a | 1410±60a |
| | PSB | 1762±318b | 163.9±26.8c | 1.48±0.28b | 0.34±0.04bc | 4593±843c | 9205±0c | 1227±300a |

Note: -Po indicates no organic P applied; Po indicates organic P applied. CK = control; PHY = phytase addition; PSB = phosphate-solubilizing bacteria. Different letters in the same column indicate significant differences (p < 0.05). Four repeats were used; the same applies below, The data are presented as the mean + standard error.

### 3.4. Mycorrhizal colonization rate and enzyme activity

The application of organic P, along with exogenous PHY and PSB, significantly influenced both the mycorrhizal colonization rate of maize roots and the enzyme activities in the rhizosphere soil (Fig 3). Compared to the no-P treatment, organic P application increased mycorrhizal colonization rate (MC), phytase, acid phosphatase (ACP), and alkaline phosphatase (ALP) activities by 63.6%, 13.8%, 12.1%, and 14.9%, respectively. Among the organic P treatments, the CK group showed significant increases in mycorrhizal colonization rate and phytase activity by 51.0% and 18.8%, respectively. The PHY treatment significantly enhanced mycorrhizal colonization, ACP, and ALP by 107.3%, 22.4%, and 14.5%. The PSB treatment led to increases in mycorrhizal colonization, phytase, ACP, and ALP by 32.7%, 14.6%, 10.9%, and 23.9%, respectively.

In the absence of sodium phytate, PHY treatment significantly increased phytase activity by 38.1% compared to CK, while ACP decreased by 19.6%. PSB treatment under the same condition significantly raised phytase and ACP by 19.9% and 14.6%, respectively. Under sodium phytate application, PHY treatment significantly enhanced mycorrhizal colonization rate and phytase activity by 55.4% and 25.4% relative to CK. PSB treatment significantly increased phytase, ACP, and ALP activities by 15.7%, 23.3%, and 20.6%, respectively (Fig 3).

### 3.5. P fractions and P pool transformation

The application of organic P, along with exogenous PHY and PSB, significantly altered the P fractions in maize rhizosphere soil (Table 2). Compared to the no-P treatment, organic P application increased the content of resin-P, NaHCO₃-Pi, NaOH-Pi, NaOH-Po, and conc.HCl-Po, while decreased the content of NaHCO₃-Po. Among them, the CK group showed significant increases in resin-P, NaHCO₃-Pi, NaOH-Pi, conc.HCl-Pi, and conc.HCl-Po by 66.2%, 15.4%, 54.3%, 16.0%, and 28.1%, respectively. The PHY treatment led to significant increases in resin-P, NaHCO₃-Pi, and NaOH-Pi by 31.1%, 11.5%, and 73.3%, respectively, while conc.HCl-Pi decreased significantly by 9.9%. The PSB treatment significantly enhanced resin-P, NaHCO₃-Pi, NaOH-Pi, and NaOH-Po by 67.8%, 14.0%, 31.1%, and 52.6%, with no significant changes in other fractions.

In the absence of sodium phytate, PHY treatment significantly increased resin-P, NaHCO₃-Pi, and conc.HCl-Pi by 90.1%, 14.4%, and 7.2%, respectively, compared to CK, while NaOH-Po decreased by 47.7%. Under the same condition, PSB treatment significantly raised NaOH-Pi and conc.HCl-Pi by 31.4% and 3.2%. When sodium phytate was applied, PHY treatment significantly increased resin-P, NaHCO₃-Pi, and NaOH-Pi by 50.0%, 10.5%, and 28.7%, respectively, relative to CK, while NaHCO₃-Po, NaOH-Po, conc.HCl-Pi, and conc.HCl-Po decreased significantly by 33.1%, 54.9%, 16.7%,

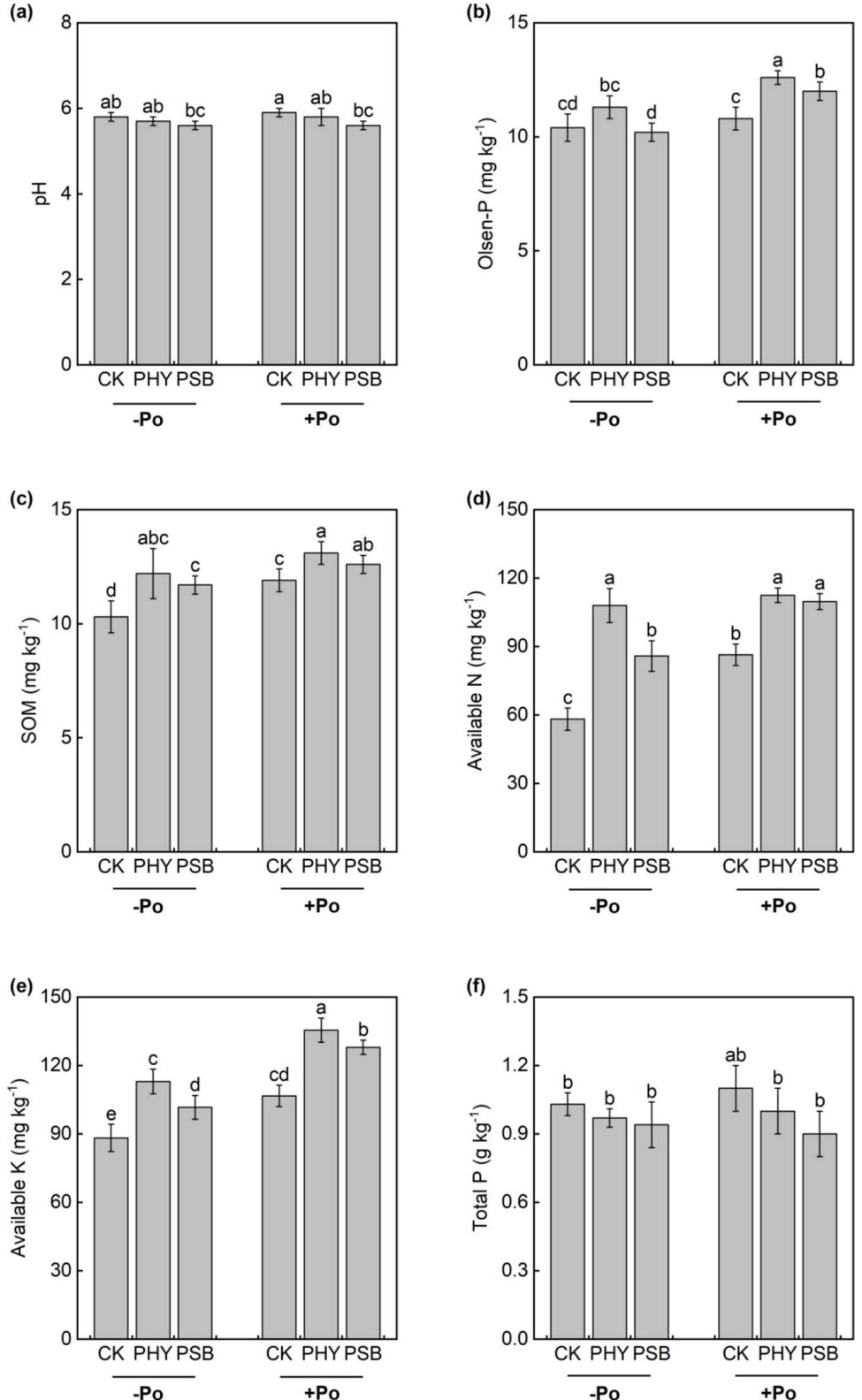

**Fig 2. Effects of exogenous addition with PHY and PSB on pH (a), Olsen-P (b), SOM (c), availzble N (d), available K (e) and total P (f) of maize with or without application organic P in low-P red soil.** Values with the same lower-case letters are not significantly different among different P

sources and exogenous addition treatments at the 5% level by the LSD. -Po represents that no organic P was applied, Po represents that organic P was applied. CK is control, PHY represents addition of phytase, PSB represents addition of phosphorus-solubilizing bacteria. the same as below.

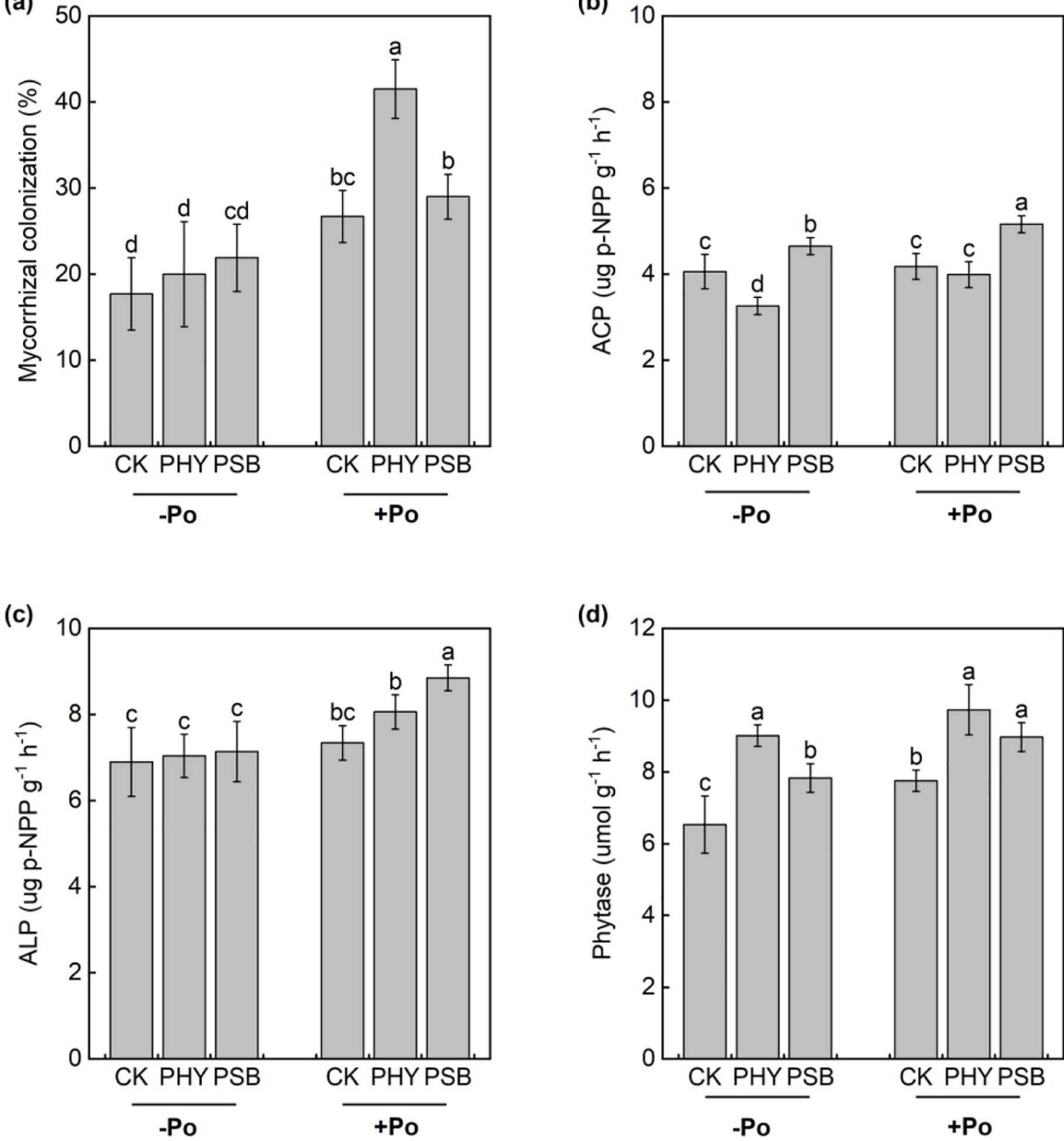

**Fig 3. Effects of exogenous addition with PHY and PSB on mycorrhizal colonization (a), ACP (b), ALP (c) and phytase (d) of maize with or without application organic P in low-P red soil.** Values with the same lower-case letters are not significantly different among different P sources and exogenous addition treatments at the 5% level by the LSD. -Po represents that no organic P was applied, Po represents that organic P was applied. CK is control, PHY represents addition of phytase, PSB represents addition of phosphorus-solubilizing bacteria. the same as below.

**Table 2.** Effects of exogenous addition with PHY and PSB on P fractions of maize rhizosphere soil with or without application organic P in low-P red soil.

| Treatments | | Resin-P (mg kg⁻¹) | NaHCO₃-Pi (mg kg⁻¹) | NaHCO₃-Po (mg kg⁻¹) | NaOH-Pi (mg kg⁻¹) | NaOH-Po (mg kg⁻¹) | 1M HCl-Pi (mg kg⁻¹) | conc.HCl-Pi (mg kg⁻¹) | conc.HCl-Po (mg kg⁻¹) | Residual P (mg kg⁻¹) |
|---|---|---|---|---|---|---|---|---|---|---|
| -Po | CK | 1.5±0.4d | 54.1±2.9d | 16.7±2.2a | 93±9.7e | 59.8±12.9ab | 13.6±1.5a | 174.8±7.2c | 98.9±10.7b | 91.9±4.8a |
| | PHY | 2.9±0.5bc | 61.8±3bc | 11±2.3b | 106.6±3.4e | 31.5±9.2c | 15.8±1.3a | 187.4±14.8ab | 95.6±5.5b | 90.6±1.7a |
| | PSB | 1.9±0.4d | 56.8±4 cd | 14.9±1.9a | 122.2±10.5d | 43.7±8bc | 15.5±2.2a | 180.5±23.7ab | 92.3±9.4b | 91.8±0.9a |
| +Po | CK | 2.5±0.4c | 62.4±4.7bc | 16±2.3a | 143.5±14.4c | 76.5±19.4a | 16±2.6a | 202.7±14.5a | 126.6±19.4a | 92.1±2.9a |
| | PHY | 3.7±0.2a | 68.9±5.2a | 10.7±1.9b | 184.7±6.8a | 34.5±8.5c | 16.7±2.1a | 168.9±4.4c | 102.8±9.9b | 92.3±5.9a |
| | PSB | 3.2±0.3ab | 64.8±3.4ab | 13.4±1.7ab | 160.3±9.1b | 66.7±14.3a | 15.3±0.6a | 180.5±23.7ab | 113.7±23.4ab | 90.7±0.9a |

Note: -Po indicates no organic P applied; Po indicates organic P applied. CK = control; PHY = phytase addition; PSB = phosphate-solubilizing bacteria. Different letters in the same column indicate significant differences (p < 0.05). Four repeats were used; the same applies below, The data are presented as the mean + standard error.

and 18.8%. PSB treatment under sodium phytate significantly enhanced resin-P and NaOH-Pi by 27.8% and 11.7%, respectively.

The application of organic P, along with exogenous PHY and PSB, significantly influenced P pool transformations in maize rhizosphere soil (Fig 4). Compared to the no-P treatment, organic P application increased the content of labile P pool, moderately-labile P pool, organic P pool, and inorganic P pool. Among them, the CK group showed significant increases in the labile P pool, moderately-labile P pool, non-labile P pool, organic P pool, and inorganic P pool by 12.0%, 41.8%, 15.3%, 16.4%, and 26.7%, respectively. The PHY treatment significantly enhanced the labile P pool, moderately-labile P pool, and inorganic P pool by 10.1%, 53.3%, and 18.3%. The PSB treatment led to significant increases in the labile P pool, moderately-labile P pool, organic P pool, and inorganic P pool by 10.5%, 33.6%, 17.3%, and 12.5%, respectively (Fig 4).

In the absence of sodium phytate, both PHY and PSB treatments significantly increased the inorganic P pool by 11.1% and 11.8%, respectively, compared to CK. Under sodium phytate application, PHY and PSB treatments significantly enhanced the non-labile P pool by 13.7% and 8.7%, and the organic P pool by 22.8% and 8.6%, respectively, relative to CK (Fig 4).

### 3.6. Correlation analysis and redundancy analysis

Maize root morphological traits were closely associated with agronomic performance and P uptake (Fig 5). Redundancy analysis (RDA) was employed to examine the relationships between root characteristics (explanatory variables) and agronomic traits and P uptake (response variables) (Fig 5a). In the RDA biplot, the angles between vectors indicate their correlations: acute angles represent positive correlations, obtuse angles negative correlations, and right angles no correlation. The projection of qualitative variable centroids onto explanatory vectors further illustrates their relationships.

The RDA model indicated that root characteristics explained 58.6% of the total variation in agronomic traits and P uptake, with RDA1 and RDA2 accounting for 55.3% and 3.3%, respectively. Root branching showed particularly high explanatory power. Specifically, root length and branch number were positively correlated with dry matter accumulation. Root tips, root surface area, and root volume were positively associated with P uptake, but negatively correlated with average root diameter.

Similarly, soil physicochemical properties and enzyme activities were strongly linked to soil available P (Fig 5b). An RDA model with soil properties and enzymes as explanatory variables and Olsen-P along with labile P fractions as response variables revealed that 70.3% of the variation in available P was explained by these factors. RDA1 and RDA2

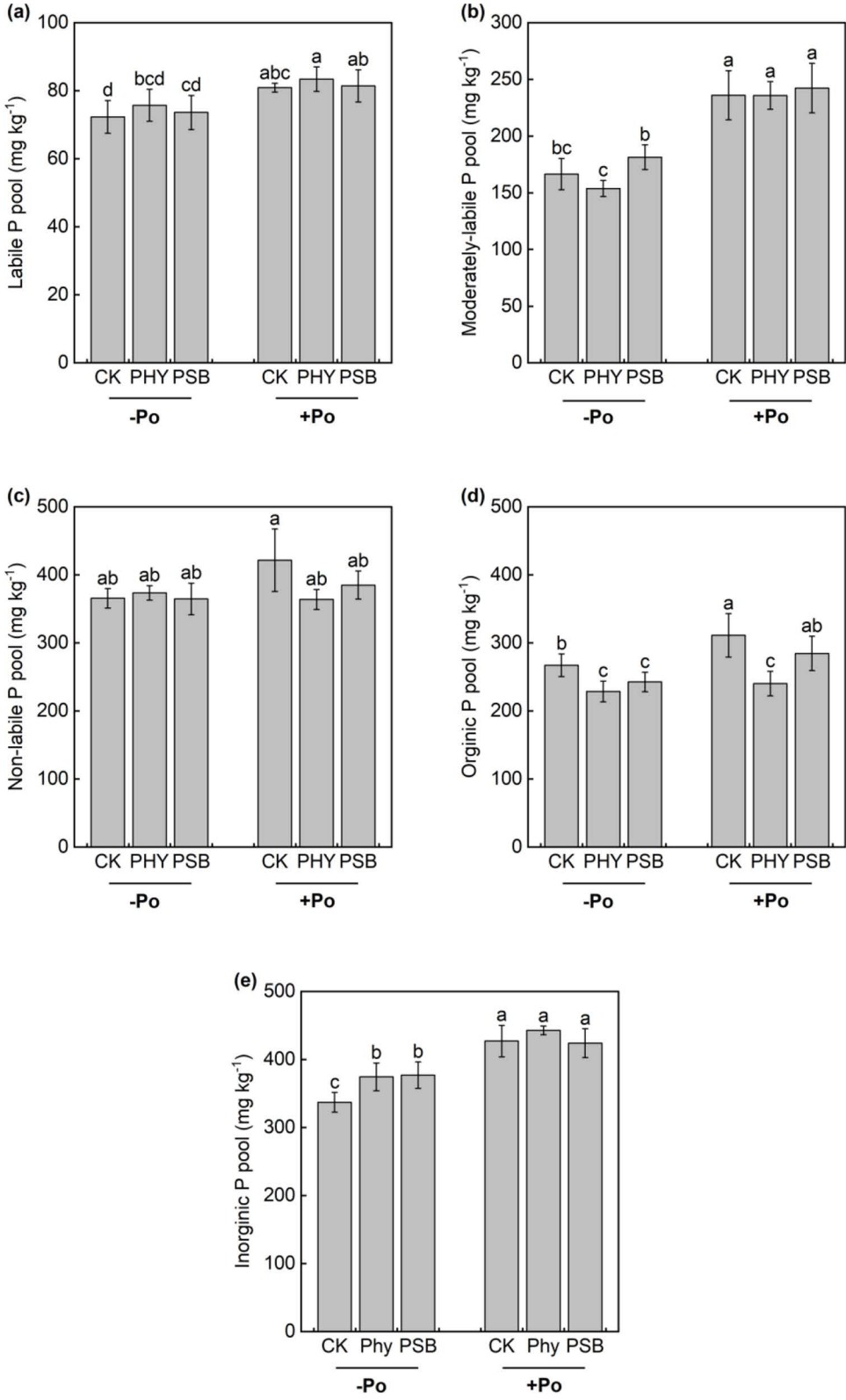

**Fig 4. Effects of exogenous addition with PHY and PSB on labile P pool (a), moderately-labile P pool (b), non-labile P pool (c), organic P pool (d) and inorganic P pools (d) in rhizosphere soil of maize with or without application organic P in low-P red soil.** Values with the same

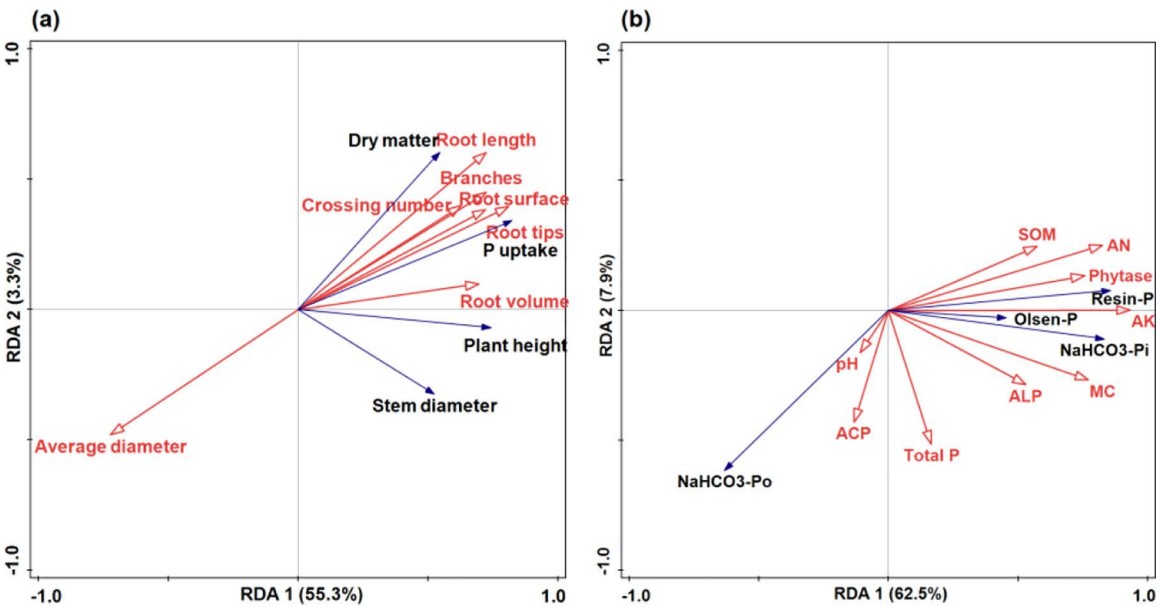

**Fig 5. Correlations between the root characteristics and P uptake, agronomic traits (a), soil physical and chemical indexes, enzyme activity and available P (b) by the redundancy analysis.** The red arrow indicates the explanatory variable, and the black arrow indicates the response variable. MC represents mycorrhizal colonization, ACP represents acid phosphatase. ALP represents alkaline phosphatase, AN represents available nitrogen, AK represents available potassium. the same as below.

accounted for 62.5% and 7.9% of this variation, respectively. Available potassium and acid phosphatase (ACP) exhibited particularly strong explanatory power. Olsen-P, resin-P, and NaHCO₃-Pi showed positive correlations with phytase, available potassium, mycorrhizal colonization (MC), and alkaline phosphatase (ALP), but were negatively correlated with NaHCO₃-Po.

The Mantel test revealed significant relationships between plant/soil variables and P fractions/pools in maize rhizosphere soil (Fig 6a). Agronomic traits showed significant correlations with NaOH-Pi. Root characteristics were significantly correlated with resin-P, NaHCO₃-Pi, NaHCO₃-Po, NaOH-Pi, NaOH-Po, and residual P. Similarly, physicochemical properties exhibited significant associations with resin-P, NaHCO₃-Pi, NaHCO₃-Po, NaOH-Pi, NaOH-Po, and residual P. Enzyme activity was significantly correlated with resin-P, NaHCO₃-Pi, and NaOH-Pi.

Regarding P pools, the labile P pool showed positive relationships with resin-P, NaHCO₃-Pi, and NaOH-Pi, but negative correlations with NaHCO₃-Po and residual P. The moderately-labile P pool was positively associated with NaOH-Pi. The non-labile P pool demonstrated positive correlations with conc.HCl-Pi and conc.HCl-Po. The organic P pool was positively related to NaOH-Po and conc.HCl-Po, but negatively correlated with resin-P, NaHCO₃-Pi, and 1M HCl-Pi. The inorganic P pool showed positive associations with resin-P, NaHCO₃-Pi, and NaOH-Pi, but a negative relationship with NaHCO₃-Po.

Spearman's correlation analysis further indicated that agronomic traits, root characteristics, physicochemical properties, and enzyme activity significantly influenced P fractions and pools (Fig 6b). Nearly all measured indices—except average

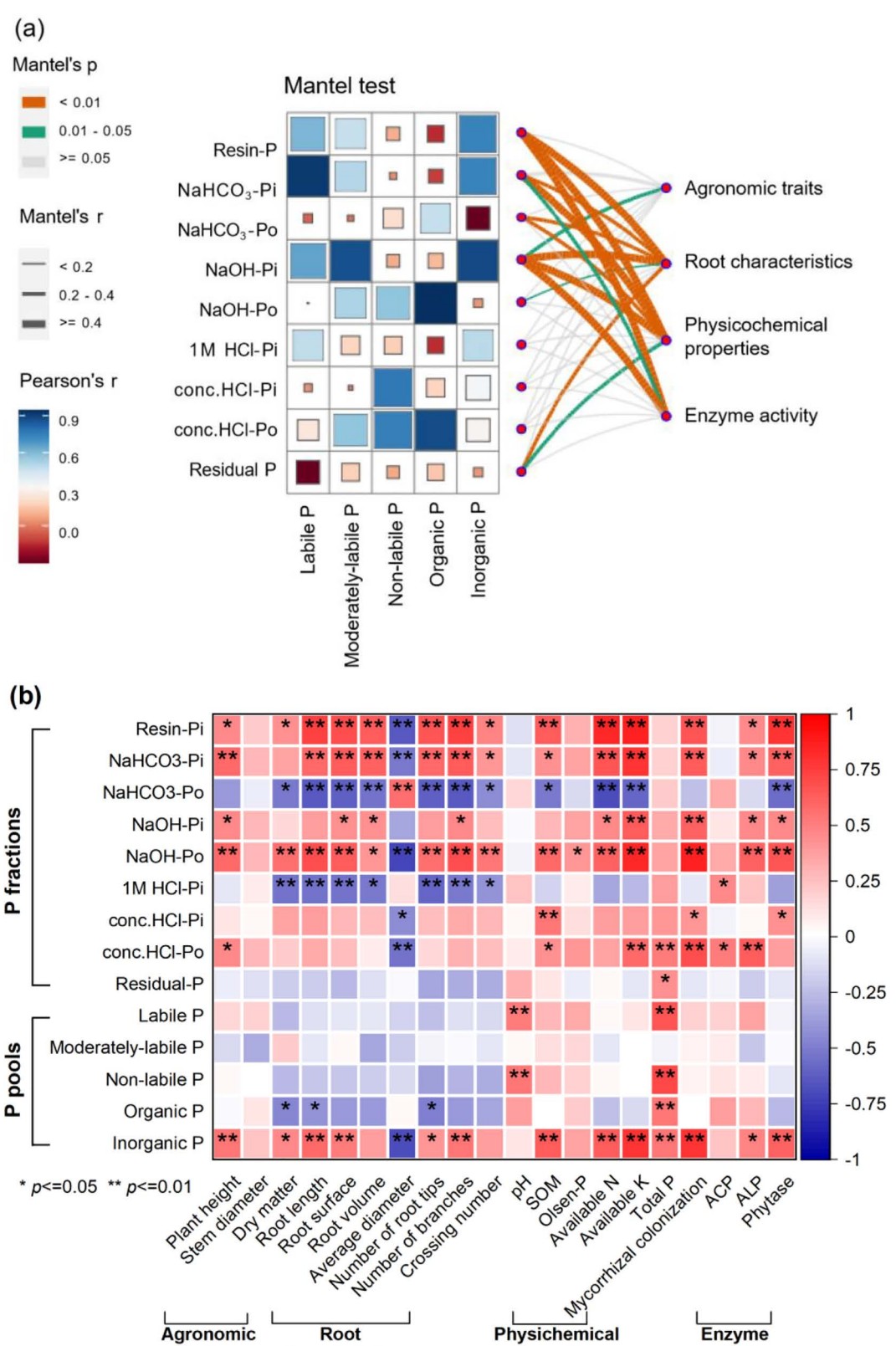

**Fig 6. Mantel analysis (a) and correlation analysis (b) of P fractions, P pool and agronomic traits, root characteristics, soil physical and chemical, enzyme activity in rhizosphere soil of maize in low-P red soil.**

root diameter—showed positive correlations with resin-P, NaHCO$_3$-Pi, NaOH-Pi, NaOH-Po, and the inorganic P pool. Conversely, these variables were generally negatively correlated with NaHCO$_3$-Po and 1M HCl-Pi.

## 4. Discussion

### 4.1. Crop growth and P uptake

Numerous studies have confirmed that inoculation with PSB or direct application of phosphatases can significantly improve P availability and crop P uptake in low-P soils [48–50]. Consistent with previous findings, our results demonstrate that the application of organic P increased both dry matter accumulation and P uptake in maize, irrespective of phytase or PSB amendments. This indicates that crop-soil interactions can effectively mobilize insoluble organic P even in highly weathered red soils under low-P conditions. Furthermore, the exogenous application of both PHY and PSB significantly enhanced maize dry matter production and P uptake. This improvement can be attributed to elevated P-related enzyme activities and enhanced P mobilization in the rhizosphere, leading to greater P acquisition [51]. These results align with earlier reports confirming the beneficial effects of microbial and enzymatic interventions on P nutrition and crop productivity in phosphorus-deficient environments [51].

### 4.2. Root characteristic

Adaptive changes in root morphology play a decisive role in plant P acquisition [52,53]. In this study, the application of organic P increased root surface area, root volume, and root branching, while reducing average root diameter compared to the control. These results indicate that organic P amendments stimulate morphological changes in roots that facilitate P absorption in low-P soil, consistent with previous findings [26]. Furthermore, the exogenous application of either PHY or PSB significantly enhanced multiple root morphological parameters—including root length, surface area, volume, tip number, branching, and root crossings—while decreasing average diameter, regardless of P fertilization. This suggests that microbial activity plays a key role in rhizosphere P mobilization. By secreting P-solubilizing enzymes or directly influencing phytate availability, these treatments enhance microbial and phosphatase activities, thereby inducing root architectural changes that improve P acquisition efficiency [54]. These findings help elucidate the root morphological mechanisms through which exogenous PHY and PSB promote P use efficiency.

Supporting these results, Ramesh et al. [22] reported that PSB inoculation improved plant height, root and shoot biomass, root volume, surface area, and P uptake in soybean and wheat. Similar improvements in growth parameters following inoculation with P-solubilizing and phytate-mineralizing bacteria have been widely documented [55]. In contrast, Yu et al. [56] observed that inoculation with *Bacillus* cereus W9 did not significantly enhance plant growth or P uptake, highlighting strain- and context-dependent effectiveness. Microbial activities also alter the composition of root exudates and release metabolites that further influence P cycling and availability in the rhizosphere [57,58].

### 4.3. Soil physichemistry properties

Exogenous inoculation of PSB significantly enhances soil P availability and promotes plant P uptake by regulating soil physicochemical properties and enzyme activities, offering a promising strategy to improve fertilizer efficiency and reduce P loss in agriculture. This study demonstrates that organic P application improved soil fertility in the maize rhizosphere, with treatment effects (CK, PHY, PSB) further modulated by the presence or absence of sodium phytate. In the absence of sodium phytate, the PHY treatment was particularly effective at increasing available P, suggesting its suitability for scenarios targeting P deficiency. In contrast, when sodium phytate was applied, the PSB treatment showed stronger synergistic improvements in multiple nutrients—especially available nitrogen.

Compared to the no-P control, organic P application significantly enhanced the physicochemical properties and nutrient supply capacity of the rhizosphere soil. These findings underscore the central role of organic P in improving the maize

rhizosphere environment. The PHY treatment notable increases the content of Olsen-P and available potassium, indicating its capacity to directly solubilize insoluble P and facilitate potassium release. The PSB treatment significantly elevated organic matter, Olsen-P, available nitrogen, and available potassium, suggesting broader microbial-mediated nutrient transformations, possibly through shifts in microbial community structure and the secretion of hydrolytic enzymes such as phosphatases and proteases [36,59].

Our results demonstrate that PHY increased organic matter, available P, and available potassium without sodium phytate. The marked rise in available P with PHY likely reflects the high specificity of phytase toward native soil organic P, such as phytate. In contrast, PSB—likely reliant on organic acid secretion and indirect microbial processes—exhibited a lower native P activation capacity (47.8% increase in Olsen-P). Under sodium phytate amendment, the Olsen-P increases were less pronounced for both PHY and PSB, possibly due to competitive inhibition at enzymatic sites or complexation of sodium phytate with metal ions (e.g., $Ca^{2+}$, $Fe^{3+}$), reducing bioavailability. Conversely, available nitrogen increases were more substantial under sodium phytate, indicating enhanced nitrogen mineralization possibly driven by improved P availability and P-dependent enzymatic activities, reflecting coupled N and P cycling. Notably, PSB inoculation under sodium phytate significantly reduced soil pH (by 5.1%), likely due to microbial secretion of organic acids (e.g., citric, oxalic acids), which contribute to P solubilization and alter nutrient availability, e.g., promoting potassium release from minerals, thereby supporting the significant increase (20.0%) in available potassium.

Additionally, both PHY and PSB treatments strengthened the arbuscular mycorrhizal symbiosis and increased the activities of P-cycle-related enzymes in the rhizosphere. The PHY treatment showed the highest increases in mycorrhizal colonization (107.3%), acid phosphatase (22.4%), and alkaline phosphatase (14.5%), suggesting a dual role: direct enzymatic P mobilization and potential induction of plant symbiotic signaling. The PSB treatment achieved more balanced enhancements across all measured enzymes and mycorrhization, with the highest increase in alkaline phosphatase (23.9%), indicating a broader microbial ecological function involving diverse P forms and microbial consortia [39]. These results align with earlier studies reporting increased phosphatase activities and improved P availability following *Bacillus* inoculation [37,60]. Similarly, PSB inoculation in Camellia oleifera was shown to elevate labile P pools (e.g., $CaCl_2$-P, citrate-P, enzyme-P), reduce stable HCl-P, and enhance phosphatase activity through modulation of the rhizobacterial community, thereby promoting P availability and plant nutrient uptake [61].

### 4.4. Soil P fractions and P bioavailability

In our results, the exogenous PSB application reduced the content of insoluble or non-labile inorganic P fractions while increasing labile P pools in maize rhizosphere soil. This can be attributed to two main mechanisms: (1) secretion of organic acids that solubilize mineral-bound P, and (2) enhanced enzymatic mineralization of organic P through elevated phosphatase and phytase activities. The effect was particularly pronounced under organic P amendment, aligning with previous reports that combined application of organic P sources and PSB promotes P flux into more available fractions [23]. The observed increases in $NaHCO_3$-Pi and NaOH-Pi in this study likely reflect enhanced mineralization of organic P ($NaHCO_3$-Po and NaOH-Po), driven by microbial enzymatic activity. This interpretation is supported by other studies linking inoculation with increased phosphatase and phytase activities [57,62]. Bünemann [10] noted that up to 60% of soil organic P can be hydrolyzed by phosphatases, with phytases playing a dominant role—consistent with the decline in $NaHCO_3$-Po and NaOH-Po pools following inoculation.

Mineralization of organic P is largely mediated by phosphatases, which hydrolyze phosphate monoesters into plant-available inorganic P [15,60]. Acid phosphatase (ACP) originates mainly from plant roots and microorganisms [36,63], while alkaline phosphatase (ALP) is primarily of microbial origin [64]. PSB contribute to this process by producing phytases and phosphatases that catalyze the release of P from organic compounds [65,66]. For example, Qu et al. [32] showed that phytase addition enhanced short-term organic P availability in wetland soils by converting non-labile organic

P into more labile forms. Similarly, Song et al. [67] reported that exogenous phytase increased the proportion of labile and moderately labile organic P while facilitating the conversion of non-labile organic P to bioavailable inorganic P.

In this study, exogenous phytase application primarily reduced the organic P pool while increasing labile P fractions, confirming its direct role in mineralizing organic P into plant-available forms. These results also support the concept that PSB indirectly promote P transformation through stimulation of enzymatic activity in the rhizosphere. These insights provide a theoretical basis for developing novel fertilization strategies aimed at improving soil P availability and use efficiency through microbial and enzymatic interventions.

### 4.5. Relationships between soil available P and Soil P fractions

Soil physicochemical properties and enzyme activities are closely linked to P availability, while root morphological traits play a crucial role in crop growth, P accumulation, and the modulation of soil P fractions. Fertilization practices further influence P bioavailability by altering soil properties and root development (Fig 6a) [9,22]. In this study, redundancy analysis (RDA) revealed that soil physicochemical and enzymatic variables collectively explained 70.3% of the variation in available P content. Similarly, root morphological characteristics accounted for 58.6% of the variation in soil P fractions. Specifically, Olsen-P, resin-P, and $NaHCO_3$-Pi showed positive correlations with phytase activity, available potassium, mycorrhizal colonization (MC), and alkaline phosphatase (ALP), but were negatively correlated with $NaHCO_3$-Po. Root length and branching were positively associated with dry matter accumulation, while root tips, root surface area, and root volume correlated positively with P uptake. Consistent with previous studies, modifications in root system architecture and improvements in rhizosphere soil physicochemical properties are recognized as key mechanisms influencing plant phosphorus acquisition. The enhancement of phosphorus uptake in maize is strongly associated with adaptive changes in root morphological traits, particularly increases in root length and surface area [36–40,52].

Furthermore, nearly all measured agronomic traits (e.g., plant height, dry matter), root characteristics (e.g., root length, surface area, volume), physicochemical properties (e.g., SOM, available N and K), and enzyme activities (e.g., ALP, phytase) exhibited positive correlations with resin-Pi, $NaHCO_3$-Pi, NaOH-Pi, NaOH-Po, and the inorganic P pool. In contrast, these variables were negatively correlated with $NaHCO_3$-Po and 1M HCl-Pi. The sole exception was average root diameter, which did not show a consistent relationship with P fractions. These results indicate that the combined application of PHY and PSB enhances soil P activation and maize P uptake by modifying root system architecture and improving the rhizosphere environment. Additionally, PSB inoculation contributed to broader improvements in soil fertility, supporting more efficient P cycling and plant nutrient acquisition [64,65].

## 5. Conclusions

The combined application of organic P with exogenous PHY and PSB significantly enhanced maize agronomic traits, root morphology, rhizosphere soil properties, and P availability. Organic P application exerted a more pronounced effect than the sole addition of phytase or PSB. Exogenous phytase and PSB further improved root architectural development, rhizosphere enzyme activities, and soil physicochemical conditions, facilitating the mineralization of organic P, enhancing soil P availability, and ultimately promoting P uptake and dry matter accumulation in maize. This study demonstrates that both phytase and PSB are effective in increasing P supply and improving maize growth in low-P red soils, regardless of organic P amendment. The significant improvements in P uptake and plant development underscore the potential of integrating microbial and enzymatic strategies into P management practices to achieve more sustainable and efficient agricultural fertilization.

### Author contributions

**Conceptualization:** Long Zhou, Faming Zhang.

**Data curation:** Long Zhou.

**Formal analysis:** Long Zhou.

**Funding acquisition:** Long Zhou.

**Investigation:** Long Zhou, Litao Zhu.

**Methodology:** Long Zhou.

**Software:** Long Zhou.

**Validation:** Faming Zhang.

**Writing – original draft:** Long Zhou.

**Writing – review & editing:** Long Zhou, Faming Zhang.

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
