## [Decision Letter · Decision Letter 0]

21 Jan 2026

PONE-D-25-59259Addition of phytase and phosphate-solubilizing bacteria to mediated P activation in maize rhizosphere soil and P uptake by maize in low-phosphorus red soilPLOS One

Dear Dr. Zhou,

Thank you for submitting your manuscript to PLOS ONE. After careful consideration, we feel that it has merit but does not fully meet PLOS ONE’s publication criteria as it currently stands. Therefore, we invite you to submit a revised version of the manuscript that addresses the points raised during the review process.

If applicable, we recommend that you deposit your laboratory protocols in protocols.io to enhance the reproducibility of your results. Protocols.io assigns your protocol its own identifier (DOI) so that it can be cited independently in the future. For instructions see: https://journals.plos.org/plosone/s/submission-guidelines#loc-laboratory-protocols. Additionally, PLOS ONE offers an option for publishing peer-reviewed Lab Protocol articles, which describe protocols hosted on protocols.io. Read more information on sharing protocols at . Additionally, PLOS ONE offers an option for publishing peer-reviewed Lab Protocol articles, which describe protocols hosted on protocols.io. Read more information on sharing protocols at https://plos.org/protocols?utm_medium=editorial-email&utm_source=authorletters&utm_campaign=protocols..

We look forward to receiving your revised manuscript.

Kind regards,

Rachid Bouharroud

Academic Editor

PLOS One

Journal Requirements:

“Yunnan Science and Technology Plan Project (202405AI350229), Yunnan Open University High Quality Research Project (2025G07)”

“The authors have declared that no competing interests exist. I have read the journal's policy and the authors of this manuscript have the following competing interests.”

5. We note that your Data Availability Statement is currently as follows: All relevant data are within the manuscript and in Supporting Information files.

6. We note you have included a table to which you do not refer in the text of your manuscript. Please ensure that you refer to Table 1 and 2 in your text; if accepted, production will need this reference to link the reader to the Table.

Reviewers' comments:

Reviewer's Responses to Questions

**Comments to the Author**

1. Is the manuscript technically sound, and do the data support the conclusions?

Reviewer #1: Yes

Reviewer #2: Yes

2. Has the statistical analysis been performed appropriately and rigorously? 

Reviewer #1: Yes

Reviewer #2: Yes

3. Have the authors made all data underlying the findings in their manuscript fully available?

Reviewer #1: Yes

Reviewer #2: Yes

4. Is the manuscript presented in an intelligible fashion and written in standard English?

Reviewer #1: Yes

Reviewer #2: No

5. Review Comments to the Author

Reviewer #1: Comments

The paper addresses an important topic related to phytase and phosphate-solubilizing bacteria for mediating P activation in maize rhizosphere soil and improving P uptake by maize in low-phosphorus red soil. The manuscript is well structured and rich in information. However, several minor revisions are needed, and one of the most important issues concerns the replicability of the results, since the number of repetitions is low. Overall, the document merits publication after these minor revisions.

• Please add line numbers to facilitate comment referencing.

Abstract

• Please italicize Bacillus in “phosphate-solubilizing Bacillus”.

Introduction

• In the sentence “Representative genera such as bacillus, pseudomonas, and rhizobium enhance P availability”, please capitalize the first litter of genus names and italicize them: Bacillus, Pseudomonas, and Rhizobium.

• Last paragraph: “phosphate-solubilizing bacteria (Bacillus velezensis) (PSB)” Bacillus velezensis is a species name and must always be italicized.

Materials and methods

• “The maize (Zea mays L.) cultivar…” Zea mays must be italicized.

• “a single colony of B. velezensis from a -80 °C stock” B. velezensis must be italicized.

• “to prepare a high-density seed culture.” This expression is unclear. Do you mean a high-concentration bacterial suspension? Please clarify.

• “inoculated with or without Bacillus velezensis” Use B. velezensis, italicized.

• Replication issue: “6 treatments × 4 replicates = 24 pots” is statistically low, especially for same plante like Zea mays. Even if 3–5 seeds were sown per pot, the replication number remains low and may affect experimental reproducibility. Please address or justify this.

• “Bacterial inoculation was performed by applying 50 mL of B. velezensis suspension (≈10¹⁰ CFU·mL⁻¹)”: B. velezensis must be italicized. And the concentration 10¹⁰ CFU·mL⁻¹ is very high; most studies use 10⁸ CFU·mL⁻¹ (or 10⁸ CFU·g⁻¹ of soil). Please justify your choice.

• Mycorrhizal colonization: Please explain how colonization was quantified (clearing, staining method, microscopic scoring method, etc.).

Results

• Table 1: Add a table title.

• Table 1 and Table 2: Add the explanation under the tables: “-Po indicates no organic P applied; Po indicates organic P applied. CK = control; PHY = phytase addition; PSB = phosphate-solubilizing bacteria.”

Discussion :

The long general paragraph on P availability, PSM functions, and soil enzymes belongs to the Introduction, not the Discussion. For exemeple :

• “P is an essential nutrient for plant growth, yet its availability in soil is often limited. Only a small fraction of soil P exists in water-soluble forms that are readily absorbable by crops [44]. The transformation of P within the soil plays a critical role in its uptake and utilization by plants [45]. PSMs which are ubiquitous in agricultural soils, serve as key drivers of the soil P cycle—particularly in the transformation of organic P [10]. These microorganisms facilitate not only the mineralization of organic and microbial P but also the solubilization of insoluble inorganic P, thereby enhancing plant-available P [10]. Soil enzyme activities are widely recognized as early and sensitive indicators of soil responses to perturbations such as microbial inoculation, and reflect broader ecosystem functioning [37,46]. Notably, soil microorganisms can enhance P availability through the production of phytase and related enzymes, which hydrolyze organic P compounds such as phytate [46]. Numerous studies have confirmed that inoculation with PSB or direct application of phosphatases can significantly improve P availability and crop P uptake in low-P soils [47-49]”.

Please move it to the Introduction or rewrite it briefly to focus on interpreting your findings. You may start instead with: “Our results demonstrate that the application of organic P increased…”

• Root characteristic : “Adaptive changes in root morphology play a decisive role in plant P acquisition [51,52]. Increases in root length, surface area, and diameter enhance the capacity of roots to explore soil and absorb P nutrients [53,54]. Due to the low mobility and availability of P in soil, plants largely rely on morphological adaptations—such as increased root length, expanded surface area, and reduced average diameter—to improve P uptake [34,53]. Traits such as adventitious root development, lateral root proliferation, and root hair density further contribute significantly to the acquisition of soil P [55,56].”

The paragraph describing general concepts of root morphology and P acquisition belongs to the Introduction, not the Discussion. Please shorten or move it, and focus on interpreting your own root data.

• Delete “(Fig. 2)” in the Discussion; figures should be referenced mainly in Results.

• Sentences such as “ The PHY treatment led to notable increases in Olsen- P (11.5%) and available potassium (19.9%), indicating its capacity to directly solubilize insoluble P and facilitate potassium release. The PSB treatment significantly elevated organic matter (7.7%), Olsen-P (17.6%), available nitrogen (27.7%), and available potassium (25.9%)” directly repeat Results. Please rewrite to avoid redundancy and interpret results instead of repeating them.

• “….Without sodium phytate, PHY increased organic matter, available P, and available potassium by 18.4%, 85.7%, and 28.0%, respectively, compared to CK” directly repeat Results

• The paragraph beginning with “PSB play a crucial role in P transformation within the rhizosphere, enhancing the bioavailability of otherwise inaccessible P forms and promoting plant nutrient uptake [1]. These microorganisms activate insoluble inorganic P through acidification (e.g., H⁺ release) and carboxylate excretion, while also mineralizing organic P via increased phosphatase and phytase activities [64–66]. For instance, Ramesh et al. demonstrated that PSB inoculation significantly increased soil available P and improved rice growth [57]. Similarly, Chang et al. [20] reported that PSB inoculation increased the labile P pool by 9.2% on average, while moderately labile and non-labile P pools decreased by 6.9% and 5.4%, respectively.” is general background and belongs in the introduction. Please move it or rewrite it briefly to relate it directly to your findings.

Reviewer #2: Title: It could be made applied....for example "Phytase and PSB application improves P availability and Uptake by Maize in Red soil"

Abstract; It should be very precise. General statements should be avoided. It is mentioned here that all the parameters are significantly affected by PHY and PSB, but when we look into results it varies....secondly different parameters are mentioned that they are affected significantly, but by whom its missing.

Introduction: Its very precise and to the point. The last objective mentioned should be carefully reviewed, whether its addressed during study?

Material and Methods: Its well narrated and well explained with rigorous methods, reasonable treatments and and replicates. However sample size for agronomic parameters is not visible. The data analysis is rigorously conducted with the help of standard softwares...and logical approaches.

Results:The results are comprehensively written and well equipped with tables and figures. However some results are written in generalized form and not specified. General declaration of significant results is misleadingin some cases,it should be specified.

Similarly if the difference lies in percentages, that should be not be mentioned in significant differences.

A tricky point is that the tilte of the study is focusing on PHY and PSB but in results and abstract the focus is on addition of Organic P. It should be clarified.

Some examples of generalized statements not supported by data presented....

1. The application of organic P, along with exogenous PSB and PHY, significantly influenced maize root morphological traits (Tab. 1). .....when we look the table it is not so.....

2. Compared to the no-P treatment, the application of organic P significantly enhanced several physicochemical properties of maize rhizosphere soil, increasing pH, SOM, Olsen-P, available nitrogen, and available ......when we explore fig 1 ...pH is not significantly affected.......

3.The application of organic P, along with exogenous PHY and PSB, significantly altered the P fractions in maize rhizosphere soil (Tab. 2). Compared to the no-P treatment, organic P application increased resin-P, NaHCO₃-Pi, NaOH-Pi, NaOH-Po, and conc.HCl-Po by 55.0%, 13.7%, 52.9%, .........all the P fractions were not altered significantly....

One anamoly is that in text Table 3 is mentioned but thats not visible in actual....transformations in maize rhizosphere soil (Tab. 3). Compared to the no-P treatment, organic P application

Discussion: Its well narrated with logics and references ......however latest reference is 4-5 years old.....Addition of latest references is suggested.....

6. PLOS authors have the option to publish the peer review history of their article (what does this mean?). If published, this will include your full peer review and any attached files.). If published, this will include your full peer review and any attached files.

.

Reviewer #1: No

Reviewer #2: **Yes:**Aftab AfzalAftab Afzal

---

## [Author Response · Author response to Decision Letter 1]

11 Feb 2026

Dear Reviewers:

Thank you for spending your valuable time on our manuscript, we have carefully checked the full text as recommended by the reviewers, and revise the manuscript according to every suggestion made by the reviewers. In order to ensure that the main idea can be maintained after the revision of the manuscript, we have systematically sorted out the manuscript. At the same time, we have also proofread the contents in the manuscript one by one (such as the paper format, references, etc.).

The following is an item-by-item reply to the suggestions of the reviewers.

Please add line numbers to facilitate comment referencing.

Response: Thanks to the reviewers for their suggestions, we have added line numbers to the manuscript.

Abstract

Please italicize Bacillus in “phosphate-solubilizing Bacillus”.

Response: The full text has been checked. The Bacillus species are italicized, and the descriptions of other genera are uniformly italicized with the first letter capitalized.

Introduction

In the sentence “Representative genera such as bacillus, pseudomonas, and rhizobium enhance P availability”, please capitalize the first litter of genus names and italicize them: Bacillus, Pseudomonas, and Rhizobium.

Response: Thank you to the reviewers. We have followed their suggestions and capitalized the first letter of genus names and italicized them.

Last paragraph: “phosphate-solubilizing bacteria (Bacillus velezensis) (PSB)” Bacillus velezensis is a species name and must always be italicized.

Response: The manuscript has been revised according to the reviewers' suggestions, with “Bacillus velezens” consistently being presented in italics throughout.

Materials and methods

“The maize (Zea mays L.) cultivar…” Zea mays must be italicized.

Response: “Zea mays” has been revised to be italicized and the entire text has been proofread.

“a single colony of B. velezensis from a -80 °C stock” B. velezensis must be italicized.

Response: The species B. velezensis has been adjusted to italic according to the suggestions of the reviewers.

“to prepare a high-density seed culture.” This expression is unclear. Do you mean a high-concentration bacterial suspension? Please clarify.

Response: Yes, thanks to the reviewer's suggestion. As the editor pointed out, it mean “a high-concentration bacterial suspension”. We have made the revisions in the manuscript.

“inoculated with or without Bacillus velezensis” Use B. velezensis, italicized.

Response: The species B. velezensis has been uniformly formatted in italics as per the reviewers' suggestions.

Replication issue: “6 treatments × 4 replicates = 24 pots” is statistically low, especially for same plante like Zea mays. Even if 3–5 seeds were sown per pot, the replication number remains low and may affect experimental reproducibility. Please address or justify this.

Response: Thanks to the suggestions and reminders from the reviewers, in this study, each treatment was indeed conducted 4 times with repetition, which might be sufficient compared to the conventional 3 repetitions. However, as the reviewers pointed out, 4 repetitions might still be too low. This also reminds us that in future studies, we will draw lessons from this experience and increase the number of repetitions in the experiments.

“Bacterial inoculation was performed by applying 50 mL of B. velezensis suspension (≈10¹⁰ CFU·mL⁻¹)”: B. velezensis must be italicized. And the concentration 10¹⁰ CFU·mL⁻¹ is very high; most studies use 10⁸ CFU·mL⁻¹ (or 10⁸ CFU·g⁻¹ of soil). Please justify your choice.

Response: Thank you to the reviewers for your meticulous and professional work. We have now uniformly converted B. velezensis to italic. After verification of the original processing, it was found that the B. velezensis suspension was approximately 108 CFU·mL⁻¹. Before use, we had diluted the suspension, and this was also in accordance with the usage standards in conventional research. We have revised the manuscript accordingly.

Mycorrhizal colonization: Please explain how colonization was quantified (clearing, staining method, microscopic scoring method, etc.).

Response: Thanks to the reviewers' suggestions, we have clarified and revised the manuscript. Mycorrhizal colonization is carried out using the microscopic staining examination method.

Results

Table 1: Add a table title.

Response: Thanks to the careful review and feedback from the experts, the title of Table 1 has been supplemented.

Table 1 and Table 2: Add the explanation under the tables: “-Po indicates no organic P applied; Po indicates organic P applied. CK = control; PHY = phytase addition; PSB = phosphate-solubilizing bacteria.”

Response: Thanks to the suggestions from the reviewers, we have added the abbreviations of each treatment as notes under Table 1 and Table 2 according to their recommendations.

Discussion :

The long general paragraph on P availability, PSM functions, and soil enzymes belongs to the Introduction, not the Discussion. For exemeple :

“P is an essential nutrient for plant growth, yet its availability in soil is often limited. Only a small fraction of soil P exists in water-soluble forms that are readily absorbable by crops [44]. The transformation of P within the soil plays a critical role in its uptake and utilization by plants [45]. PSMs which are ubiquitous in agricultural soils, serve as key drivers of the soil P cycle—particularly in the transformation of organic P [10]. These microorganisms facilitate not only the mineralization of organic and microbial P but also the solubilization of insoluble inorganic P, thereby enhancing plant-available P [10]. Soil enzyme activities are widely recognized as early and sensitive indicators of soil responses to perturbations such as microbial inoculation, and reflect broader ecosystem functioning [37,46]. Notably, soil microorganisms can enhance P availability through the production of phytase and related enzymes, which hydrolyze organic P compounds such as phytate [46]. Numerous studies have confirmed that inoculation with PSB or direct application of phosphatases can significantly improve P availability and crop P uptake in low-P soils [47-49]”.

Please move it to the Introduction or rewrite it briefly to focus on interpreting your findings. You may start instead with: “Our results demonstrate that the application of organic P increased…”

Response: Thank you to the reviewers. We have rewritten the discussion section. For details, please refer to the revision mode in the manuscript.

Root characteristic : “Adaptive changes in root morphology play a decisive role in plant P acquisition [51,52]. Increases in root length, surface area, and diameter enhance the capacity of roots to explore soil and absorb P nutrients [53,54]. Due to the low mobility and availability of P in soil, plants largely rely on morphological adaptations—such as increased root length, expanded surface area, and reduced average diameter—to improve P uptake [34,53]. Traits such as adventitious root development, lateral root proliferation, and root hair density further contribute significantly to the acquisition of soil P [55,56].”

The paragraph describing general concepts of root morphology and P acquisition belongs to the Introduction, not the Discussion. Please shorten or move it, and focus on interpreting your own root data.

Response: Thank you to the reviewers. We have rewritten the discussion section. For details, please refer to the revision mode in the manuscript.

Delete “(Fig. 2)” in the Discussion; figures should be referenced mainly in Results.

Response: Thank you to the reviewers. We have followed their suggestions and removed "(Fig. 2)" from the discussion section. We have also revised similar expressions in the following text.

Sentences such as “ The PHY treatment led to notable increases in Olsen- P (11.5%) and available potassium (19.9%), indicating its capacity to directly solubilize insoluble P and facilitate potassium release. The PSB treatment significantly elevated organic matter (7.7%), Olsen-P (17.6%), available nitrogen (27.7%), and available potassium (25.9%)” directly repeat Results. Please rewrite to avoid redundancy and interpret results instead of repeating them.

Response: Thanks to the reviewer, I couldn't agree more with the suggestion you made and we have further organized this part of the content to avoid repetition of discussion content and results. The details of the modifications can be found in the revised document.

“….Without sodium phytate, PHY increased organic matter, available P, and available potassium by 18.4%, 85.7%, and 28.0%, respectively, compared to CK” directly repeat Results

Response: Thank you all for your reviews. We have revised this section. The details of the revisions can be found in the revised document.

The paragraph beginning with “PSB play a crucial role in P transformation within the rhizosphere, enhancing the bioavailability of otherwise inaccessible P forms and promoting plant nutrient uptake [1]. These microorganisms activate insoluble inorganic P through acidification (e.g., H⁺ release) and carboxylate excretion, while also mineralizing organic P via increased phosphatase and phytase activities [64–66]. For instance, Ramesh et al. demonstrated that PSB inoculation significantly increased soil available P and improved rice growth [57]. Similarly, Chang et al. [20] reported that PSB inoculation increased the labile P pool by 9.2% on average, while moderately labile and non-labile P pools decreased by 6.9% and 5.4%, respectively.” is general background and belongs in the introduction. Please move it or rewrite it briefly to relate it directly to your findings.

Response: Thanks to the suggestions from the reviewers, we have moved this part to an appropriate position in the introduction and made overall adjustments to the citation of relevant references throughout the text.

Once again, we would like to express our gratitude to the reviewers for spending their valuable time on our paper and for providing practical suggestions. We fully agree with the reviewers' suggestions and have made every effort to revise accordingly. Additionally, after the overall revision, we also revised the text format proposed by the editorial department, reorganized the references, and supplemented the author information, Data availability statement, Funding, and Competing interests, etc. Since the reference version is in PDF format, there may be some inaccuracies in the revision format. We kindly ask the editors and reviewers to please understand. The other revision contents can be found in the revised version of the manuscript.

---

## [Editor Report · Decision Letter 1]

4 Mar 2026

PONE-D-25-59259R1Addition of phytase and phosphate-solubilizing bacteria to mediated P activation in maize rhizosphere soil and P uptake by maize in low-phosphorus red soilPLOS One

Dear Dr. Zhang,

Thank you for submitting your manuscript to PLOS ONE. After careful consideration, we feel that it has merit but does not fully meet PLOS ONE’s publication criteria as it currently stands. Therefore, we invite you to submit a revised version of the manuscript that addresses the points raised during the review process.

If applicable, we recommend that you deposit your laboratory protocols in protocols.io to enhance the reproducibility of your results. Protocols.io assigns your protocol its own identifier (DOI) so that it can be cited independently in the future. For instructions see: https://journals.plos.org/plosone/s/submission-guidelines#loc-laboratory-protocols. Additionally, PLOS ONE offers an option for publishing peer-reviewed Lab Protocol articles, which describe protocols hosted on protocols.io. Read more information on sharing protocols at . Additionally, PLOS ONE offers an option for publishing peer-reviewed Lab Protocol articles, which describe protocols hosted on protocols.io. Read more information on sharing protocols at https://plos.org/protocols?utm_medium=editorial-email&utm_source=authorletters&utm_campaign=protocols..

We look forward to receiving your revised manuscript.

Kind regards,

Rachid Bouharroud

Academic Editor

PLOS One

**Journal Requirements:**

**Additional Editor Comments:**

Dear

The comments from reviewer 2 have not been addressed. Please ensure you review all feedback before submitting your revised manuscript.

Good luck

---

## [Author Response · Author response to Decision Letter 2]

7 Mar 2026

Dear Reviewers:

Thank you for spending your valuable time on our manuscript, we have carefully checked the full text as recommended by the reviewers, and revise the manuscript according to every suggestion made by the reviewers. In order to ensure that the main idea can be maintained after the revision of the manuscript, we have systematically sorted out the manuscript. At the same time, we have also proofread the contents in the manuscript one by one (such as the paper format, references, etc.).

The following is an item-by-item reply to the suggestions of the reviewers.

Please add line numbers to facilitate comment referencing.

Response: Thanks to the reviewers for their suggestions, we have added line numbers to the manuscript.

Abstract

Please italicize Bacillus in “phosphate-solubilizing Bacillus”.

Response: The full text has been checked. The Bacillus species are italicized, and the descriptions of other genera are uniformly italicized with the first letter capitalized.

Introduction

In the sentence “Representative genera such as bacillus, pseudomonas, and rhizobium enhance P availability”, please capitalize the first litter of genus names and italicize them: Bacillus, Pseudomonas, and Rhizobium.

Response: Thank you to the reviewers. We have followed their suggestions and capitalized the first letter of genus names and italicized them.

Last paragraph: “phosphate-solubilizing bacteria (Bacillus velezensis) (PSB)” Bacillus velezensis is a species name and must always be italicized.

Response: The manuscript has been revised according to the reviewers' suggestions, with “Bacillus velezens” consistently being presented in italics throughout.

Materials and methods

“The maize (Zea mays L.) cultivar…” Zea mays must be italicized.

Response: “Zea mays” has been revised to be italicized and the entire text has been proofread.

“a single colony of B. velezensis from a -80 °C stock” B. velezensis must be italicized.

Response: The species B. velezensis has been adjusted to italic according to the suggestions of the reviewers.

“to prepare a high-density seed culture.” This expression is unclear. Do you mean a high-concentration bacterial suspension? Please clarify.

Response: Yes, thanks to the reviewer's suggestion. As the editor pointed out, it mean “a high-concentration bacterial suspension”. We have made the revisions in the manuscript.

“inoculated with or without Bacillus velezensis” Use B. velezensis, italicized.

Response: The species B. velezensis has been uniformly formatted in italics as per the reviewers' suggestions.

Replication issue: “6 treatments × 4 replicates = 24 pots” is statistically low, especially for same plante like Zea mays. Even if 3–5 seeds were sown per pot, the replication number remains low and may affect experimental reproducibility. Please address or justify this.

Response: Thanks to the suggestions and reminders from the reviewers, in this study, each treatment was indeed conducted 4 times with repetition, which might be sufficient compared to the conventional 3 repetitions. However, as the reviewers pointed out, 4 repetitions might still be too low. This also reminds us that in future studies, we will draw lessons from this experience and increase the number of repetitions in the experiments.

“Bacterial inoculation was performed by applying 50 mL of B. velezensis suspension (≈10¹⁰ CFU·mL⁻¹)”: B. velezensis must be italicized. And the concentration 10¹⁰ CFU·mL⁻¹ is very high; most studies use 10⁸ CFU·mL⁻¹ (or 10⁸ CFU·g⁻¹ of soil). Please justify your choice.

Response: Thank you to the reviewers for your meticulous and professional work. We have now uniformly converted B. velezensis to italic. After verification of the original processing, it was found that the B. velezensis suspension was approximately 108 CFU·mL⁻¹. Before use, we had diluted the suspension, and this was also in accordance with the usage standards in conventional research. We have revised the manuscript accordingly.

Mycorrhizal colonization: Please explain how colonization was quantified (clearing, staining method, microscopic scoring method, etc.).

Response: Thanks to the reviewers' suggestions, we have clarified and revised the manuscript. Mycorrhizal colonization is carried out using the microscopic staining examination method.

Results

Table 1: Add a table title.

Response: Thanks to the careful review and feedback from the experts, the title of Table 1 has been supplemented.

Table 1 and Table 2: Add the explanation under the tables: “-Po indicates no organic P applied; Po indicates organic P applied. CK = control; PHY = phytase addition; PSB = phosphate-solubilizing bacteria.”

Response: Thanks to the suggestions from the reviewers, we have added the abbreviations of each treatment as notes under Table 1 and Table 2 according to their recommendations.

Discussion :

The long general paragraph on P availability, PSM functions, and soil enzymes belongs to the Introduction, not the Discussion. For exemeple :

“P is an essential nutrient for plant growth, yet its availability in soil is often limited. Only a small fraction of soil P exists in water-soluble forms that are readily absorbable by crops [44]. The transformation of P within the soil plays a critical role in its uptake and utilization by plants [45]. PSMs which are ubiquitous in agricultural soils, serve as key drivers of the soil P cycle—particularly in the transformation of organic P [10]. These microorganisms facilitate not only the mineralization of organic and microbial P but also the solubilization of insoluble inorganic P, thereby enhancing plant-available P [10]. Soil enzyme activities are widely recognized as early and sensitive indicators of soil responses to perturbations such as microbial inoculation, and reflect broader ecosystem functioning [37,46]. Notably, soil microorganisms can enhance P availability through the production of phytase and related enzymes, which hydrolyze organic P compounds such as phytate [46]. Numerous studies have confirmed that inoculation with PSB or direct application of phosphatases can significantly improve P availability and crop P uptake in low-P soils [47-49]”.

Please move it to the Introduction or rewrite it briefly to focus on interpreting your findings. You may start instead with: “Our results demonstrate that the application of organic P increased…”

Response: Thank you to the reviewers. We have rewritten the discussion section. For details, please refer to the revision mode in the manuscript.

Root characteristic : “Adaptive changes in root morphology play a decisive role in plant P acquisition [51,52]. Increases in root length, surface area, and diameter enhance the capacity of roots to explore soil and absorb P nutrients [53,54]. Due to the low mobility and availability of P in soil, plants largely rely on morphological adaptations—such as increased root length, expanded surface area, and reduced average diameter—to improve P uptake [34,53]. Traits such as adventitious root development, lateral root proliferation, and root hair density further contribute significantly to the acquisition of soil P [55,56].”

The paragraph describing general concepts of root morphology and P acquisition belongs to the Introduction, not the Discussion. Please shorten or move it, and focus on interpreting your own root data.

Response: Thank you to the reviewers. We have rewritten the discussion section. For details, please refer to the revision mode in the manuscript.

Delete “(Fig. 2)” in the Discussion; figures should be referenced mainly in Results.

Response: Thank you to the reviewers. We have followed their suggestions and removed "(Fig. 2)" from the discussion section. We have also revised similar expressions in the following text.

Sentences such as “ The PHY treatment led to notable increases in Olsen- P (11.5%) and available potassium (19.9%), indicating its capacity to directly solubilize insoluble P and facilitate potassium release. The PSB treatment significantly elevated organic matter (7.7%), Olsen-P (17.6%), available nitrogen (27.7%), and available potassium (25.9%)” directly repeat Results. Please rewrite to avoid redundancy and interpret results instead of repeating them.

Response: Thanks to the reviewer, I couldn't agree more with the suggestion you made and we have further organized this part of the content to avoid repetition of discussion content and results. The details of the modifications can be found in the revised document.

“….Without sodium phytate, PHY increased organic matter, available P, and available potassium by 18.4%, 85.7%, and 28.0%, respectively, compared to CK” directly repeat Results

Response: Thank you all for your reviews. We have revised this section. The details of the revisions can be found in the revised document.

The paragraph beginning with “PSB play a crucial role in P transformation within the rhizosphere, enhancing the bioavailability of otherwise inaccessible P forms and promoting plant nutrient uptake [1]. These microorganisms activate insoluble inorganic P through acidification (e.g., H⁺ release) and carboxylate excretion, while also mineralizing organic P via increased phosphatase and phytase activities [64–66]. For instance, Ramesh et al. demonstrated that PSB inoculation significantly increased soil available P and improved rice growth [57]. Similarly, Chang et al. [20] reported that PSB inoculation increased the labile P pool by 9.2% on average, while moderately labile and non-labile P pools decreased by 6.9% and 5.4%, respectively.” is general background and belongs in the introduction. Please move it or rewrite it briefly to relate it directly to your findings.

Response: Thanks to the suggestions from the reviewers, we have moved this part to an appropriate position in the introduction and made overall adjustments to the citation of relevant references throughout the text.

Once again, we would like to express our gratitude to the reviewers for spending their valuable time on our paper and for providing practical suggestions. We fully agree with the reviewers' suggestions and have made every effort to revise accordingly. Additionally, after the overall revision, we also revised the text format proposed by the editorial department, reorganized the references, and supplemented the author information, Data availability statement, Funding, and Competing interests, etc. Since the reference version is in PDF format, there may be some inaccuracies in the revision format. We kindly ask the editors and reviewers to please understand. The other revision contents can be found in the revised version of the manuscript.

---

## [Editor Report · Decision Letter 2]

25 Mar 2026

PONE-D-25-59259R2Addition of phytase and phosphate-solubilizing bacteria to mediated P activation in maize rhizosphere soil and P uptake by maize in low-phosphorus red soilPLOS One

Dear Dr. Zhang,

Thank you for submitting your manuscript to PLOS ONE. After careful consideration, we feel that it has merit but does not fully meet PLOS ONE’s publication criteria as it currently stands. Therefore, we invite you to submit a revised version of the manuscript that addresses the points raised during the review process.

If applicable, we recommend that you deposit your laboratory protocols in protocols.io to enhance the reproducibility of your results. Protocols.io assigns your protocol its own identifier (DOI) so that it can be cited independently in the future. For instructions see: https://journals.plos.org/plosone/s/submission-guidelines#loc-laboratory-protocols. Additionally, PLOS ONE offers an option for publishing peer-reviewed Lab Protocol articles, which describe protocols hosted on protocols.io. Read more information on sharing protocols at . Additionally, PLOS ONE offers an option for publishing peer-reviewed Lab Protocol articles, which describe protocols hosted on protocols.io. Read more information on sharing protocols at https://plos.org/protocols?utm_medium=editorial-email&utm_source=authorletters&utm_campaign=protocols..

As the corresponding author, your ORCID iD is verified in the submission system and will appear in the published article. PLOS supports the use of ORCID, and we encourage all coauthors to register for an ORCID iD and use it as well. Please encourage your coauthors to verify their ORCID iD within the submission system before final acceptance, as unverified ORCID iDs will not appear in the published article. *Only* the individual author can complete the verification step; PLOS staff the individual author can complete the verification step; PLOS staff *cannot* verify ORCID iDs on behalf of authors.verify ORCID iDs on behalf of authors.

We look forward to receiving your revised manuscript.

Kind regards,

Rachid Bouharroud

Academic Editor

PLOS One

Journal Requirements:

Additional Editor Comments:

Dear

The comments from reviewer 2 have not been addressed. Please ensure you review all feedback before submitting your revised manuscript.

Good luck

---

## [Author Response · Author response to Decision Letter 3]

27 Mar 2026

Dear Reviewers:

Thank you for spending your valuable time on our manuscript, we have carefully checked the full text as recommended by the reviewers, and revise the manuscript according to every suggestion made by the reviewers. In order to ensure that the main idea can be maintained after the revision of the manuscript, we have systematically sorted out the manuscript. At the same time, we have also proofread the contents in the manuscript one by one (such as the paper format, references, etc.).

The following is the detailed response to the suggestions made by the two reviewers.

For reviewer 1

Please add line numbers to facilitate comment referencing.

Response: Thanks to the reviewers for their suggestions, we have added line numbers to the manuscript.

Abstract

Please italicize Bacillus in “phosphate-solubilizing Bacillus”.

Response: The full text has been checked. The Bacillus species are italicized, and the descriptions of other genera are uniformly italicized with the first letter capitalized.

Introduction

In the sentence “Representative genera such as bacillus, pseudomonas, and rhizobium enhance P availability”, please capitalize the first litter of genus names and italicize them: Bacillus, Pseudomonas, and Rhizobium.

Response: Thank you to the reviewers. We have followed their suggestions and capitalized the first letter of genus names and italicized them.

Last paragraph: “phosphate-solubilizing bacteria (Bacillus velezensis) (PSB)” Bacillus velezensis is a species name and must always be italicized.

Response: The manuscript has been revised according to the reviewers' suggestions, with “Bacillus velezens” consistently being presented in italics throughout.

Materials and methods

“The maize (Zea mays L.) cultivar…” Zea mays must be italicized.

Response: “Zea mays” has been revised to be italicized and the entire text has been proofread.

“a single colony of B. velezensis from a -80 °C stock” B. velezensis must be italicized.

Response: The species B. velezensis has been adjusted to italic according to the suggestions of the reviewers.

“to prepare a high-density seed culture.” This expression is unclear. Do you mean a high-concentration bacterial suspension? Please clarify.

Response: Yes, thanks to the reviewer's suggestion. As the editor pointed out, it mean “a high-concentration bacterial suspension”. We have made the revisions in the manuscript.

“inoculated with or without Bacillus velezensis” Use B. velezensis, italicized.

Response: The species B. velezensis has been uniformly formatted in italics as per the reviewers' suggestions.

Replication issue: “6 treatments × 4 replicates = 24 pots” is statistically low, especially for same plante like Zea mays. Even if 3–5 seeds were sown per pot, the replication number remains low and may affect experimental reproducibility. Please address or justify this.

Response: Thanks to the suggestions and reminders from the reviewers, in this study, each treatment was indeed conducted 4 times with repetition, which might be sufficient compared to the conventional 3 repetitions. However, as the reviewers pointed out, 4 repetitions might still be too low. This also reminds us that in future studies, we will draw lessons from this experience and increase the number of repetitions in the experiments.

“Bacterial inoculation was performed by applying 50 mL of B. velezensis suspension (≈10¹⁰ CFU·mL⁻¹)”: B. velezensis must be italicized. And the concentration 10¹⁰ CFU·mL⁻¹ is very high; most studies use 10⁸ CFU·mL⁻¹ (or 10⁸ CFU·g⁻¹ of soil). Please justify your choice.

Response: Thank you to the reviewers for your meticulous and professional work. We have now uniformly converted B. velezensis to italic. After verification of the original processing, it was found that the B. velezensis suspension was approximately 108 CFU·mL⁻¹. Before use, we had diluted the suspension, and this was also in accordance with the usage standards in conventional research. We have revised the manuscript accordingly.

Mycorrhizal colonization: Please explain how colonization was quantified (clearing, staining method, microscopic scoring method, etc.).

Response: Thanks to the reviewers' suggestions, we have clarified and revised the manuscript. Mycorrhizal colonization is carried out using the microscopic staining examination method.

Results

Table 1: Add a table title.

Response: Thanks to the careful review and feedback from the experts, the title of Table 1 has been supplemented.

Table 1 and Table 2: Add the explanation under the tables: “-Po indicates no organic P applied; Po indicates organic P applied. CK = control; PHY = phytase addition; PSB = phosphate-solubilizing bacteria.”

Response: Thanks to the suggestions from the reviewers, we have added the abbreviations of each treatment as notes under Table 1 and Table 2 according to their recommendations.

Discussion :

The long general paragraph on P availability, PSM functions, and soil enzymes belongs to the Introduction, not the Discussion. For exemeple :

“P is an essential nutrient for plant growth, yet its availability in soil is often limited. Only a small fraction of soil P exists in water-soluble forms that are readily absorbable by crops [44]. The transformation of P within the soil plays a critical role in its uptake and utilization by plants [45]. PSMs which are ubiquitous in agricultural soils, serve as key drivers of the soil P cycle—particularly in the transformation of organic P [10]. These microorganisms facilitate not only the mineralization of organic and microbial P but also the solubilization of insoluble inorganic P, thereby enhancing plant-available P [10]. Soil enzyme activities are widely recognized as early and sensitive indicators of soil responses to perturbations such as microbial inoculation, and reflect broader ecosystem functioning [37,46]. Notably, soil microorganisms can enhance P availability through the production of phytase and related enzymes, which hydrolyze organic P compounds such as phytate [46]. Numerous studies have confirmed that inoculation with PSB or direct application of phosphatases can significantly improve P availability and crop P uptake in low-P soils [47-49]”.

Please move it to the Introduction or rewrite it briefly to focus on interpreting your findings. You may start instead with: “Our results demonstrate that the application of organic P increased…”

Response: Thank you to the reviewers. We have rewritten the discussion section. For details, please refer to the revision mode in the manuscript.

Root characteristic : “Adaptive changes in root morphology play a decisive role in plant P acquisition [51,52]. Increases in root length, surface area, and diameter enhance the capacity of roots to explore soil and absorb P nutrients [53,54]. Due to the low mobility and availability of P in soil, plants largely rely on morphological adaptations—such as increased root length, expanded surface area, and reduced average diameter—to improve P uptake [34,53]. Traits such as adventitious root development, lateral root proliferation, and root hair density further contribute significantly to the acquisition of soil P [55,56].”

The paragraph describing general concepts of root morphology and P acquisition belongs to the Introduction, not the Discussion. Please shorten or move it, and focus on interpreting your own root data.

Response: Thank you to the reviewers. We have rewritten the discussion section. For details, please refer to the revision mode in the manuscript.

Delete “(Fig. 2)” in the Discussion; figures should be referenced mainly in Results.

Response: Thank you to the reviewers. We have followed their suggestions and removed "(Fig. 2)" from the discussion section. We have also revised similar expressions in the following text.

Sentences such as “ The PHY treatment led to notable increases in Olsen- P (11.5%) and available potassium (19.9%), indicating its capacity to directly solubilize insoluble P and facilitate potassium release. The PSB treatment significantly elevated organic matter (7.7%), Olsen-P (17.6%), available nitrogen (27.7%), and available potassium (25.9%)” directly repeat Results. Please rewrite to avoid redundancy and interpret results instead of repeating them.

Response: Thanks to the reviewer, I couldn't agree more with the suggestion you made and we have further organized this part of the content to avoid repetition of discussion content and results. The details of the modifications can be found in the revised document.

“….Without sodium phytate, PHY increased organic matter, available P, and available potassium by 18.4%, 85.7%, and 28.0%, respectively, compared to CK” directly repeat Results

Response: Thank you all for your reviews. We have revised this section. The details of the revisions can be found in the revised document.

The paragraph beginning with “PSB play a crucial role in P transformation within the rhizosphere, enhancing the bioavailability of otherwise inaccessible P forms and promoting plant nutrient uptake [1]. These microorganisms activate insoluble inorganic P through acidification (e.g., H⁺ release) and carboxylate excretion, while also mineralizing organic P via increased phosphatase and phytase activities [64–66]. For instance, Ramesh et al. demonstrated that PSB inoculation significantly increased soil available P and improved rice growth [57]. Similarly, Chang et al. [20] reported that PSB inoculation increased the labile P pool by 9.2% on average, while moderately labile and non-labile P pools decreased by 6.9% and 5.4%, respectively.” is general background and belongs in the introduction. Please move it or rewrite it briefly to relate it directly to your findings.

Response: Thanks to the suggestions from the reviewers, we have moved this part to an appropriate position in the introduction and made overall adjustments to the citation of relevant references throughout the text.

Once again, we would like to express our gratitude to the reviewers for spending their valuable time on our paper and for providing practical suggestions. We fully agree with the reviewers' suggestions and have made every effort to revise accordingly. Additionally, after the overall revision, we also revised the text format proposed by the editorial department, reorganized the references, and supplemented the author information, Data availability statement, Funding, and Competing interests, etc. Since the reference version is in PDF format, there may be some inaccuracies in the revision format. We kindly ask the editors and reviewers to please understand. The other revision contents can be found in the revised version of the manuscript.

For reviewer 2

Title: It could be made applied....for example "Phytase and PSB application improves P availability and Uptake by Maize in Red soil"

Response: Thank you for the suggestions from the reviewers. The title of the manuscript has been optimized. "Phytase and phosphate-solubilizing Bacteria addition improves P availability and uptake by maize in low-phosphorus red soil"

Abstract; It should be very precise. General statements should be avoided. It is mentioned here that all the parameters are significantly affected by PHY and PSB, but when we look into results it varies....secondly different parameters are mentioned that they are affected significantly, but by whom its missing.

Response: Thank you to the reviewers. We have reorganized the abstract section and added the objects for comparing different parameters.

Introduction: Its very precise and to the point. The last objective mentioned should be carefully reviewed, whether its addressed during study?

Response: Thank you to the reviewers. We have read the abstract in full and made adjustments to some parts to ensure consistency with the research content and objectives.

Material and Methods: Its well narrated and well explained with rigorous methods, reasonable treatments and and replicates. However sample size for agronomic parameters is not visible. The data analysis is rigorously conducted with the help of standard softwares...and logical approaches.

Response: The entire experiment involved a total of 6 treatments, each replicated 4 times (a total of 24 pots), meaning each treatment was repeated 4 times. Data Processing and Statistical Analysis can be found in Section 2.4.

Results:The results are comprehensively written and well equipped with tables and figures. However some results are written in generalized form and not specified. General declaration of significant results is misleadingin some cases,it should be specified.

Similarly if the difference lies in percentages, that should be not be mentioned in significant differences.

A tricky point is that the tilte of the study is focusing on PHY and PSB but in results and abstract the focus is on addition of Organic P. It should be clarified.

Response: Thank you for the suggestions from the reviewers. This study examines the effects of exogenous addition of PHY and PSB on the growth traits and phosphorus uptake of maize, while using no phosphorus application as the control and applying organic phosphorus as the treatment. This study is based on different phosphorus application treatments and involves exogenous addition of PHY and PSB. Therefore, in the description of the manuscript, the treatment factors with significant effects are described. Please refer to the Materials and Methods section for details.

Some examples of generalized statements not supported by data presented....

1. The application of organic P, along with exogenous PSB and PHY, significantly influenced maize root morphological traits (Tab. 1). .....when we look the table it is not so.....

Response: As can be clearly seen from Table 1, the application of organophosphorus and the addition of exogenous PSB and PHY have varying degrees of impact on the root characteristics of corn, as can be observed from each table. We did not describe that all treatments significantly increased or decreased, as similar to "significantly affecting" - we believe this description is feasible.

2. Compared to the no-P treatment, the application of organic P significantly enhanced several physicochemical properties of maize rhizosphere soil, increasing pH, SOM, Olsen-P, available nitrogen, and available ......when we explore fig 1 ...pH is not significantly affected.......

Response: Thank you for the suggestions from the reviewers. We have now rechecked and revised this part. Please refer to the revised version of the document for details.

3.The application of organic P, along with exogenous PHY and PSB, significantly altered the P fractions in maize rhizosphere soil (Tab. 2). Compared to the no-P treatment, organic P application increased resin-P, NaHCO₃-Pi, NaOH-Pi, NaOH-Po, and conc.HCl-Po by 55.0%, 13.7%, 52.9%, .........all the P fractions were not altered significantly....

Response: Thank you for the suggestions from the reviewers. We have now rechecked and revised this part. Deleted the descriptions that did not show significant differences.

One anamoly is that in text Table 3 is mentioned but thats not visible in actual....transformations in maize rhizosphere soil (Tab. 3). Compared to the no-P treatment, organic P application

Response: We are grateful for the careful detection by the reviewers. Additionally, we made an error in citing the table. The description in this section should refer to Figure 4 instead of Table 3. We have rechecked and revised the citations and related descriptions at the corresponding positions in the manuscript.

Discussion: Its well narrated with logics and references ......however latest reference is 4-5 years old.....Addition of latest references is suggested.....

Response: Thank you to the reviewers. Here, we have supplemented some new references based on the actual situation. Please refer to the revised version of the document for details.

Once again, we would like to express our gratitude to the reviewers for the valuable time they spent in reviewing our manuscript! We would like to express our gratitude for your professionalism and meticulousness during the review process of the manuscript.

---

## [Editor Report · Decision Letter 3]

1 Apr 2026

Phytase and phosphate-solubilizing Bacteria addition improves P availability and uptake by maize in low-phosphorus red soil

PONE-D-25-59259R3

Dear Dr. Zhang,

We’re pleased to inform you that your manuscript has been judged scientifically suitable for publication and will be formally accepted for publication once it meets all outstanding technical requirements.

An invoice will be generated when your article is formally accepted. Please note, if your institution has a publishing partnership with PLOS and your article meets the relevant criteria, all or part of your publication costs will be covered. Please make sure your user information is up-to-date by logging into Editorial Manager at Editorial Manager® and clicking the ‘Update My Information' link at the top of the page. For questions related to billing, please contact  and clicking the ‘Update My Information' link at the top of the page. For questions related to billing, please contact billing support..

Kind regards,

Rachid Bouharroud

Academic Editor

PLOS One
---

## [Editor Report · Acceptance letter]

PONE-D-25-59259R3

PLOS One

Dear Dr. Zhang,

I'm pleased to inform you that your manuscript has been deemed suitable for publication in PLOS One. Congratulations! Your manuscript is now being handed over to our production team.

Kind regards,

on behalf of

Dr. Rachid Bouharroud

Academic Editor

PLOS One